# Regional specialization of movement encoding across the primate sensorimotor cortex

Simon Borgognon[1,2,3,4,5,15], Nicolò Macellari [1,2,3,6,15], Alexandra M. Hickey[1,2,3,4], Matthew G. Perich [7,8], Houman Javaheri[9], Rafael Ornelas-Kobayashi[1,2,3], Maude Delacombaz[1,2,3,4], Christopher Hitz[1,2,3], Florian Fallegger[10], Stéphanie P. Lacour [10], Erwan Bezard [11], Eric M. Rouiller [4], Jocelyne Bloch [1,2,3], Tomislav Milekovic [1,2,3,16] ✉, Ismael Seáñez [1,2,3,12,13,14,16] ✉ & Grégoire Courtine [1,2,3,16] ✉

The process by which the cerebral cortex generates movements to achieve different tasks remains poorly understood. Here, we leveraged the rich repertoire of well-controlled primate locomotor behaviors to study how task-specific movements are encoded across the dorsal premotor cortex (PMd), primary motor cortex (M1), and primary somatosensory cortex (S1) under naturalistic conditions. Neural population activity was confined within low-dimensional manifolds and partitioned into task-dependent and task-independent subspaces. However, the prevalence of these subspaces differed between cortical regions. PMd primarily operated within its task-dependent subspace, while S1, and to a lesser extent M1, largely evolved within their task-independent subspaces. The temporal structure of movement was encoded in the task-independent subspaces, which also dominated the PMd-to-M1 communication as the movement plans were translated into motor commands. Our results suggest that the brain utilizes different cortical regions to serialize the motor control by first performing task-specific computations in PMd to then generate task-independent commands in M1.

Moving through the environment to seek food, escape predators, and find partners determines survival and fitness of animals. This imperative pressured the primate nervous system to evolve an advanced cerebral cortex to support locomotion through diverse and rich environments[1,2]. Indeed, the acquisition of manual dexterity likely originated from the mechanisms that subserve skilled locomotion[3]. Therefore, understanding the cortical control of locomotion may lead to the discovery of fundamental principles through which the sensorimotor cortex produces movement. The diversity of locomotor tasks enables exploring the neural space during naturalistic behaviors, as opposed to the more standardized and, in part, artificial tasks typically involved in the study of manual dexterity. Moreover,

locomotion offers the advantage of combining automated movements with volitional adjustments to meet the requirements of each task[4]. Here, we aimed to exploit the naturalistic and rich repertoire of primate locomotor behaviors to uncover fundamental principles underlying the cortical control of movement.

The diversity of primate behavior involves complex interactions between many interconnected regions that transform the context of the task and sensory information into a pattern of muscle activations[5–9]. In particular, the premotor cortex is believed to extract high-level contextual information to initiate the planning of motor output, moderated by primary motor and primary somatosensory cortices[10–12]. Previous studies showed that a wide variety of tasks can be

completed by flexibly altering the activity of the same neurons and muscles[13–15]. This transformation from task-specific context to task-agnostic muscle activation implies that a gradual transformation from task-specific representations to task-independent commands takes place in the cortex[16]. We sought to understand these fundamental transformations that are the basis of motor control by comparing neuronal activity across multiple regions of the brain during the execution of various motor tasks. However, traditional experimental paradigms that studied individual neurons or isolated cortical regions[17–21] are unable to parse the interactions in this complex, multi-regional process.

Modern computational analyses applied to recordings of neural population activity have illuminated this possibility[22,23]. This approach leverages the correlated activity of interconnected neurons to provide a more comprehensive view into the mechanisms that lead to the outputs of the population[5,14,15,24]. The vast majority of neural population activity has been found to reside within a low-dimensional portion of the full neural space, called the neural manifold[15,24]. This neural manifold is an aggregate of orthogonal vectors, called the neural modes, that each capture correlated neuronal activity across the entire population of recorded neurons. Neural modes are thought to reflect intrinsic constraints defined by network connectivity[15,25]. Accordingly, the organization of neural modes can be remarkably preserved across tasks, even if the activity of individual neurons and muscles exhibits substantial changes to produce each task[14] or when neuronal activity varies across days or even years[26,27]. Maintaining most of the activity within a small number of modes can be advantageous for the cortex since this stability promotes the generation of behaviors that are resistant to noise, neuronal death, and synapse liability[28].

Modes that remain unchanged across tasks have been identified in several brain regions[29–33]. Recent studies suggest that the flexible combination of these task-independent modes within a cortical region or between multiple cortical regions may be a common mechanism supporting neural computation and driving population dynamics[14,34,35]. However, brain regions such as the dorsal premotor cortex (PMd, F2), primary motor cortex (M1, F1) and somatosensory cortex (S1, area 1) are known to contribute in unique ways to the planning[11,36], execution[37,38], and perception of movement[39]. Consequently, we predicted that the neural activity of each of these regions will organize to reflect the differing functional specializations.

We translated this prediction into three specific hypotheses. First, we hypothesized that each region has unique and distinct modes during performance of each task. Second, we hypothesized that the predominance of task-independent neural modes varies in a principled manner between higher order (PMd, fewer task-independent modes) and lower order (S1 and M1, more task-independent modes) regions. Lastly, we hypothesized that lower regions inherit their organization from higher order regions that plan the task at hand.

## Results

### Wireless multi-regional neural recordings during naturalistic behaviors

To test our hypotheses, we aimed to record neural population activity from multiple sensorimotor cortical regions of nonhuman primates (NHPs) as they performed a variety of naturalistic locomotor behaviors (Fig. 1). To record neural activity, we fabricated custom-made connectors combining three microelectrode "Utah" arrays that we implanted in hindlimb regions of left PMd, M1, and S1 in two adult female *Macaca fascicularis* monkeys (Mk-Ek and Mk-Nt). In Mk-Ek's connector, we integrated wire electrodes that were implanted into major hindlimb muscles to monitor electromyographic (EMG) activity. These custom-made connectors were mounted on the skull using a personalized anchoring system (Supplementary Fig. 1). Two wireless data transmission modules were attached on the skull-mounted connectors to broadcast the signals to external computers synchronized with video-based full-body kinematic recordings (Fig. 1a). We subsequently confirmed that the Utah arrays were implanted in the hindlimb region of PMd, M1, and S1 using cortical anatomical landmarks and analysis of recorded neural activity during isolated movements (Supplementary Fig. 2).

To capture naturalistic locomotor behavior, we designed and fabricated adaptable Plexiglas enclosures that allowed rapid transitions between five distinct locomotor tasks: a straight corridor, an unevenly-spaced horizontal ladder, a corridor with a three-step staircase, a corridor with two obstacles, and a treadmill (Fig. 1b). This platform enabled synchronized recordings of muscle activity and kinematics of the right hindlimb, and activity of neural populations from the left PMd, M1, and S1 across all five tasks within a single day while the monkeys behaved naturally without any tethered electronics (Fig. 2a–b). Analysis of muscle activity and hindlimb kinematics confirmed that each task involved specific changes to accommodate hindlimb movements to the constraints inherent to each task (Fig. 2c–f).

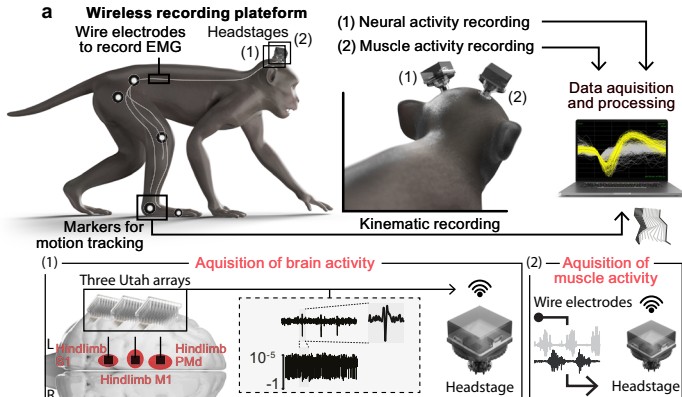

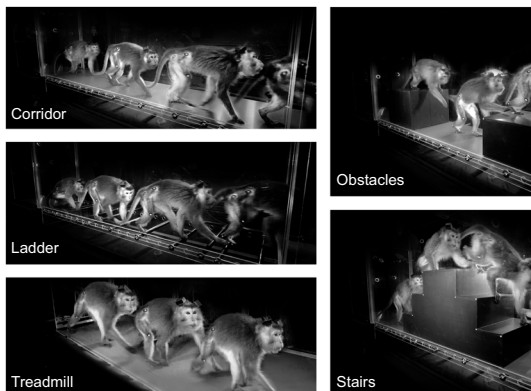

**Fig. 1 | Neurotechnological platform to capture hindlimb kinematics combined with untethered recordings of neural and muscle activity across a variety of locomotor tasks. a** We recorded kinematics of the right hindlimb using markers placed on the hindlimb anatomical landmarks. Additionally, only in Mk-Ek, we chronically implanted pairs of wire electrodes into major hindlimb muscles to record electromyographic (EMG) signals. We implanted Utah arrays into the hindlimb regions of the left dorsal premotor cortex (PMd), primary motor cortex (M1), and primary somatosensory cortex (S1) to record spiking activity of neural populations. Neural and EMG signals were broadcasted wirelessly via two wireless data transmission modules that were connected to the skull-mounted connectors during recordings. **b** We trained two monkeys (Mk-Ek and Mk-Nt) to walk across a corridor, a horizontal ladder, a staircase and over obstacles; and to walk on a treadmill moving at 3 km/h.

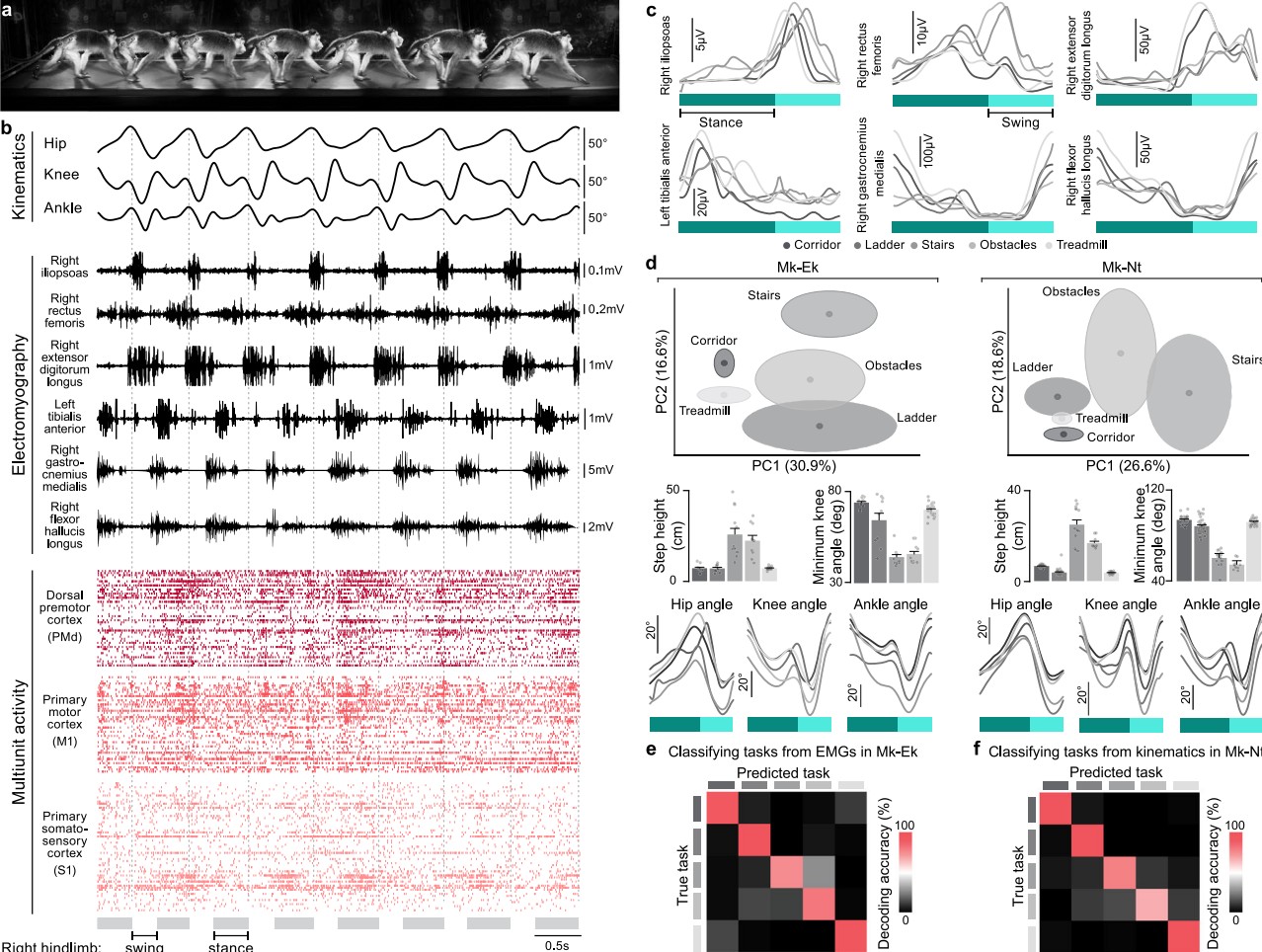

**Fig. 2 | Time-varying kinematics, electromyography (EMG), and neural population activity during walking. a** Illustrative chronophotography of 7 successive corridor steps. **b** Right hindlimb kinematics, EMG, and neural signals recorded from Mk-Ek during 5.26 seconds of walking across the corridor. Top panel: changes in hip, knee and ankle joint angles from the right hindlimb contralateral to neural recordings. Middle panel: EMG activity from six hindlimb muscles. Bottom panel: raster plots reconstructed from the multiunit activity of 160 microelectrodes distributed over three Utah arrays implanted in the hindlimb region of left PMd, M1, and S1. The gray and white spaces at the bottom indicate the duration of the stance and swing gait phases of the right hindlimb, respectively. **c** Muscle activity varies substantially across tasks. The graphs show the mean peri-gait EMG of six hindlimb muscles recorded in Mk-Ek for all five tasks. **d** Hindlimb kinematics of both monkeys differs substantially across tasks. Top: We computed 58 variables from right hindlimb kinematic recordings of each gait cycle. These variables quantified different kinematic features of monkeys' locomotor patterns (Supplementary Table 1). This dataset was arranged in a matrix with variables as the matrix columns and each

row representing one gait cycle. Data from all five locomotor tasks were pooled together in a single matrix and z-scored across columns. We then applied principal component analysis on this dataset and visualized the outcome by plotting the dataset in a space spanned by the two leading principal components (PCs). The data for each task is represented by an ellipsoid with the center and principal semi-axis as the mean and standard deviation calculated across all the gait cycles for that task. Middle: The bar plots show the mean of two variables used for the principal component analysis: the step height (number of steps: Mk-Ek: corridor: 11, ladder: 10, stairs: 12, obstacles: 11, and treadmill: 23) and minimum knee angle across steps (number of steps: Mk-Nt: corridor: 31, ladder: 37, stairs: 14, obstacles: 10, and treadmill: 25). Bottom: The graphs show the mean peri-gait hip, knee and ankle angles for all five tasks. **e** Confusion matrix reporting decoding accuracy of classifying a task from EMG envelopes in Mk-Ek. **f** Confusion matrix reporting accuracy of classifying a task from kinematic trajectories in Mk-Nt. Source data are provided as a Source Data file.

## Larger task-specific changes of neuronal responses in PMd compared to M1 and S1

We first asked whether activity of single neurons across PMd, M1 and S1 underwent comparable changes across the different tasks. To enable this comparison, each monkey performed seven sessions during which they performed all five locomotor tasks within approximately two hours. Offline spike sorting enabled the identification of putative neurons with stable action potential waveforms within each session from PMd (Mk-Ek: 114 neurons; $16.3 \pm 3.5$ per session; Mk-Nt: 131 neurons; $18.7 \pm 3.2$ per session), M1 (Mk-Ek: 147 neurons; $21.0 \pm 1.7$ per session; Mk-Nt: 88 neurons; $12.6 \pm 3.9$ per session), and S1 (Mk-Ek: 79 neurons; $11.3 \pm 1.1$ per session; Mk-Nt: 123 neurons; $17.6 \pm 2.0$ per session).

We calculated single neuron firing rates during each gait cycle, which we defined as the epoch between two consecutive foot strikes of the right hindlimb contralateral to neural recordings. We identified the stance and swing phases for each gait cycle and then time-warped these periods to 60% and 40% of the average step duration, respectively[40]. We averaged these warped trials to obtain "peri-gait" firing rates for each task separately (Fig. 3a).

Neural population activity recorded from PMd, M1, and S1 exhibited highly reproducible patterns of modulation that remained phase-locked to gait events. The overall distribution of neuronal activity across a gait cycle remained similar across tasks (Fig. 3b). Interestingly, the distribution of preferred gait phases in PMd was largely homogeneous, while M1 activity was mildly bi-modal with peaks

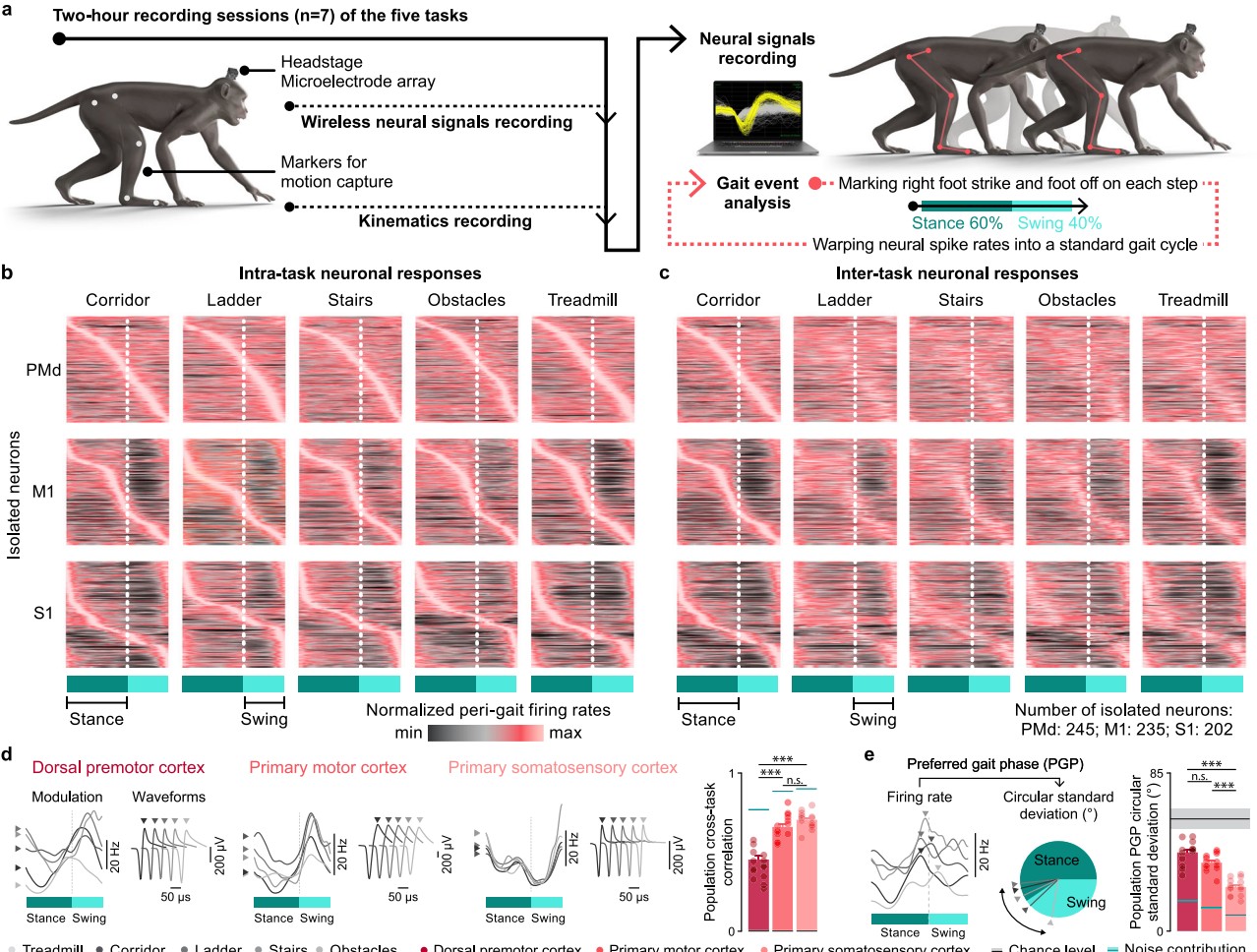

**Fig. 3 | Neural correlates of behavior vary across tasks in PMd, but stabilize in M1 and even more in S1. a** We calculated single neuron firing rates during each gait cycle, defined as the epoch between two consecutive foot strikes of the right hindlimb contralateral to neural recordings. We then identified the right hindlimb stance and swing phases for each gait cycle, and then time-warped these periods to 60% and 40% of the gait cycle. We averaged these warped trials across all gait cycles to obtain peri-gait neuronal firing rates for each task. **b** The colorplots show peri-gait firing rates of isolated neurons aligned at stance and swing onsets and sorted by their preferred gait phase on each task separately. The neurons recoded in different sessions are grouped together for visualization purposes. All statistics shown later are calculated by treating different sessions as separate data sets with no assumptions about whether the neurons recorded in different sessions are the same or different. **c** Same data as (**b**) with the neurons sorted by their preferred gait-phase in the corridor task. Note the substantial changes in neural activity, reflected by changes in gait phase tuning for each neuron. **d** Left panels show peri-gait firing rates of one representative neuron from PMd, M1, and S1. Similarity across tasks indicates that the recorded signals came from the same neuron. For every neuron, we computed the Pearson's linear correlation coefficient (R) between the activity on any two gait cycles belonging to different tasks. We then averaged these values across all gait cycle pairs in a particular task combination (10 combinations), all neurons from a cortical region, and all task combinations. The bars show the population cross-task correlation as the mean of these cross-task region-specific correlation values across all sessions and both monkeys ($n = 14$; PMd: 0.42 ± 0.03; M1: 0.67 ± 0.02; S1: 0.71 ± 0.01). Each dot shows the region-specific mean correlation value for one session of one monkey (dots on the left column, Mk-Ek; dots on the right column, Mk-Nt). Black line shows the chance level with the grey tube showing the values not significantly different from chance at $p \geq 0.05$ (chance

level: PMd: 0.0002 ± 0.009; M1: 0.0000 ± 0.0013; S1: 0.0000 ± 0.001; measured value vs. chance: PMd: $p = 0.0005$; M1: $p = 0.0005$; S1: $p = 0.0005$). Blue line shows the estimated noise contribution with light blue tube showing the values not significantly different from noise contribution at $p \geq 0.05$ (PMd: 0.768 ± 0.009; M1: 0.885 ± 0.005; S1: 0.896 ± 0.005; measured value vs. noise contribution: PMd: $p = 0.0005$; M1: $p = 0.0005$; S1: $p = 0.0005$). Population cross-task correlation was lower in PMd than in M1 and S1 (PMd vs. M1: $p = 0.0005$; PMd vs. S1: $p = 0.0005$; M1 vs. S1: $p = 0.13$). **e** Left panel shows the peri-gait firing rates for one representative neuron with its preferred gait phase (PGP) identified as the phase of maximum activity. We identified each neuron's PGP for each task and computed the circular standard deviation of PGP across tasks. The bars show the population PGP standard deviation as the mean PGP circular standard deviation across all neurons of a cortical region, all sessions and both monkeys ($n = 14$; PMd: 42.7 ± 1.7; M1: 37.1 ± 1.2; S1: 23.9 ± 1.2). Each dot shows the mean value for each session and each monkey (dots on the left column, Mk-Ek; dots on the right column, Mk-Nt). Black line shows the chance level with the grey tube showing the values not significantly different from chance at $p \geq 0.05$ (chance level: 60.33 ± 10.85; measured value vs. chance: PMd: $p = 0.0005$; M1: $p = 0.0005$; S1: $p = 0.0005$). Blue line shows the estimated noise contribution with the light blue tube showing the values not significantly different from noise contribution at $p \geq 0.05$ (noise contribution: PMd: 18.4 ± 0.9; M1: 14.5 ± 0.9; S1: 10.0 ± 0.4; measured value vs. noise contribution: PMd: $p = 0.0005$; M1: $p = 0.0005$; S1: $p = 0.0005$). Population PGP circular standard deviation trended smaller in M1 compared to PMd, and was smaller in S1 compared to M1 (PMd vs. M1: $p = 0.058$; PMd vs. S1: $p = 0.0005$; M1 vs. S1: $p = 0.0005$). Error bars: s.e.m.; n.s. $p \geq 0.05$; *** $p < 0.001$; one-sided Monte Carlo Permutation test. Source data are provided as a Source Data file.

coinciding with the foot off and foot strike gait events. S1 activity had a strong tuning to the foot strike event, which likely reflected the salient sensory stimulus emerging from the contact between the foot and the ground (Supplementary Fig. 3).

We next determined the changes in amplitude and timing of single neuron activity between tasks. To quantify these changes, we computed the mean firing rate and modulation depth of all the neurons from each region and each task. Both the firing rate and modulation depth were consistently higher in M1 and S1 compared to PMd (Mk-Ek: firing rate: M1: $27.7 \pm 2.0$ Hz; S1: $22.9 \pm 1.4$ Hz; PMd: $20.5 \pm 1.3$ Hz M1 vs. PMd: $p = 0.016$; S1 vs. PMd: $p = 0.031$; modulation depth: M1: $44.8 \pm 2.1$ Hz; S1: $45.7 \pm 2.0$ Hz; PMd: $33.3 \pm 2.0$ Hz; M1 vs. PMd: $p = 0.016$; S1 vs. PMd: $p = 0.016$; Mk-Nt: firing rate: M1: $21.0 \pm 2.1$ Hz; S1: $23.2 \pm 1.8$ Hz PMd: $14.2 \pm 1.1$ Hz; M1 vs. PMd: $p = 0.078$; S1 vs. PMd: $p = 0.031$; modulation depth: M1: $39.6 \pm 2.7$ Hz; S1: $42.2 \pm 2.7$ Hz; PMd: $26.2 \pm 1.0$ Hz; M1 vs. PMd: $p = 0.016$; S1 vs. PMd: $p = 0.016$; Wilcoxon signed rank test; Supplementary Fig. 4a).

In agreement with previous studies conducted in cats during locomotion[41,42], we found across all three cortical regions that neuronal firing rate and modulation depth modestly increased during ladder, stairs, and obstacles tasks compared to treadmill and corridor—as expected based on the requirement for better-planned and more accurate hindlimb movements compared to rhythmic walking along the corridor or treadmill (Supplementary Fig. 4b).

To capture task-specific changes in the timing of neural responses, we ordered the activity of each neuron based on their preferred gait phase, defined as the time of peak firing. We first aligned the responses during walking along the corridor, and then maintained this order when visualizing neuronal responses in the other tasks. We noticed substantial task-specific changes in the timing of neuronal responses in PMd, while the activity of neurons located in M1, and even more in S1, was comparatively more stable (Fig. 3c, Supplementary Fig. 5a). To quantify these changes, we computed the correlation coefficient (R) between the activity of any two gait cycles for each neuron, either from the same or a different task. We then averaged these values across all gait cycle pairs in a particular task combination (10 combinations) and across all neurons from a cortical region (Supplementary Fig. 5b). To calculate the region-specific mean correlation values, we averaged these R values across all pairs of different tasks. This quantification revealed that single neuron activity was less correlated across tasks in PMd than in M1 and S1 (population cross-task correlation: PMd: $0.42 \pm 0.03$; M1: $0.67 \pm 0.02$; S1: $0.71 \pm 0.01$; PMd vs. M1: $p = 0.0005$; PMd vs. S1: $p = 0.0005$ one-side Monte Carlo permutation test; Fig. 3d). We also calculated cross-task circular standard deviation of the preferred gait phases (PGP) of each neuron, and compared these values across the cortical regions. This analysis confirmed that task-specific changes in the timing of neuronal responses were larger in PMd than in M1, and were the smallest in S1 (population PGP circular standard deviation: PMd: $42.7 \pm 1.7$; M1: $37.1 \pm 1.2$; S1: $23.9 \pm 1.2$; PMd vs. M1: $p = 0.058$; PMd vs. S1: $p = 0.0005$; M1 vs. S1: $p = 0.0005$; one-sided Monte Carlo permutation test; Fig. 3e). Given the known role of PMd in movement planning[11,36,43], these results imply that PMd is a critical center for incorporating task context to flexibly redeploy motor output.

## PMd neural manifolds change substantially between tasks, but not in M1/S1

The detailed understanding of the mechanisms through which the activity of individual neurons generates muscle activation patterns to sustain movement remains incomplete[44]. Indeed, muscle activation patterns are likely orchestrated through combinations of single neuron activities that are embedded within low-dimensional neural manifolds[15]. Yet, it remains unclear whether these manifolds remain unchanged as the behavior shifts from one task to another. To address this knowledge gap, we determined the dimensionality of

neural manifolds during the production of each task, and how it relates to the dimensionality of the neural manifold when all the tasks are grouped. This analysis is bounded by two extremes. On one side, if neural population activity takes place within the same neural manifold for each task, the dimensionality of the single-task and all-task neural manifolds will be identical. On the other side, if the activity within a cortical region takes place along orthogonal neural manifolds for each task, the dimensionality of the all-task manifold will be equal to the sum of the dimensionality of the single-task manifolds. While real-life datasets likely fall between these two extremes, the proximity to either of the two signifies which type of task segregation dominates the activity of the studied neural population.

We measured the dimensionality of single- and all-task manifolds for each recording session by applying the principal component analysis (PCA) on the activity of a neural population from one cortical region and counting the number of neural modes needed to explain more than 90% of the cumulative variance, as described previously[45,46] (Fig. 4a). All the single-task neural manifolds had a low (2-4) dimensionality, regardless of the cortical region (mean single-task neural manifold dimensionality: Mk-Ek: PMd: $3.46 \pm 0.92$; M1: $3.34 \pm 0.48$; S1: $2.20 \pm 0.46$; Mk-Nt: PMd: $3.77 \pm 0.81$; M1: $2.77 \pm 0.55$; S1: $3.29 \pm 0.52$; Fig. 4b; Supplementary Fig. 6a-b). In contrast, the dimensionality of the all-task neural manifolds differed between PMd and the other two cortical regions (mean all-task neural manifold dimensionality: PMd: $8.05 \pm 0.44$; M1: $5.33 \pm 0.29$; S1: $4.78 \pm 0.44$; PMd vs. M1: $p = 0.0005$; PMd vs. S1: $p = 0.0006$; M1 vs. S1: $p = 0.30$; Wilcoxon signed rank test; Fig. 4c). In PMd, the dimensionality of all-task manifolds surpassed the dimensionality of single-task manifolds by 5.81 on average, while this difference was far smaller in M1 (2.66), and even smaller in S1 (2.11; PMd vs. M1: $p = 0.0002$; PMd vs. S1: $p = 0.0002$; M1 vs. S1: $p = 0.15$, Wilcoxon signed rank test; Fig. 4d). These differences imply that neural activity for different tasks may be compartmentalized within distinct neural manifolds in PMd, while neural activity in S1 and, to a lesser extent M1, appears to take place along similarly oriented manifolds. These results remained qualitatively unchanged for different cumulative variance thresholds used to define neural manifolds (Supplementary Fig. 6c).

Note that the design of the behavioral tasks and the processing used to derive multiunit spike rates from the neural population activity influences the dimensionality of the manifolds[47]. To estimate instantaneous spike rate from a timeseries of discrete neuronal spikes, the timeseries is convolved with a "smoothing" function (e.g., a Gaussian function). Longer tasks and narrower smoothing will inherently contain more task-unrelated variance, thus resulting in higher dimensionality estimates. Conversely, wide smoothing applied on neural recordings from short tasks may erase relevant neural activity. The maximum measurable dimensionality can be estimated by dividing the task duration by the width of the smoothing filter. With the mean gait cycle duration of 942 ms and the width of our smoothing filter of 128 ms, the single-task dimensionality ceiling is estimated to be 7.4. The measured single-task manifold dimensionalities (2-4) are substantially lower than the limit, thus indicating that the task design and the processing did not influence the obtained results. As expected, increasing the dimensionality ceiling by making our Gaussian smoothing filter half as wide did not substantially change the results, while reducing the dimensionality ceiling by making the filter twice as wide compressed the dimensionalities (Supplementary Fig. 6d).

Nevertheless, we could not exclude the possibility that the differences between single- and all-task neural manifolds arose from a single task associated with neural activity inhabiting a manifold largely orthogonal to the remaining single-task neural manifolds. To rule out this possibility, we computed the relative contribution of each task toward the dimensionality of the all-task manifold. We estimated the dimensionality of all-but-one task neural manifolds for all combinations of tasks. Excluding one task from the datasets modestly reduced

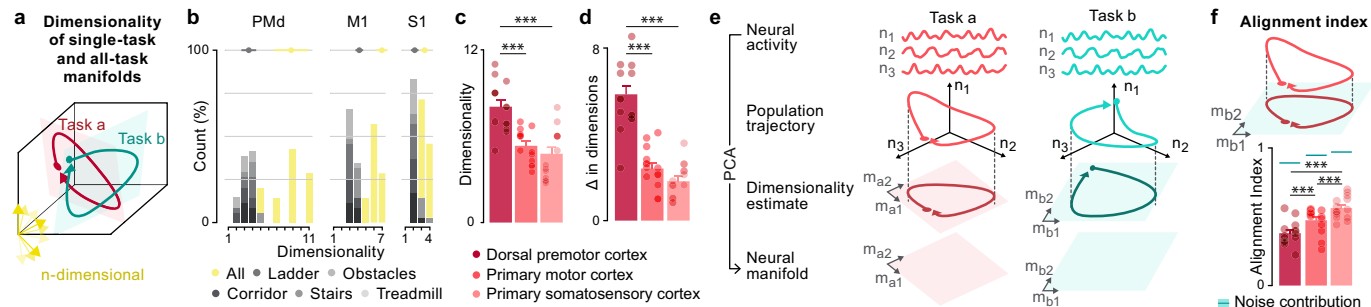

**Fig. 4 | Single-task neural manifolds are very dissimilar in PMd, moderately dissimilar in M1, and similar in S1. a.\** The dimensionality of a dataset was estimated as the number of neural modes needed to explain more than 90% of the cumulative variance. The panel illustrates the process of comparing the dimensionality of single-task and all-task neural manifolds. Here, two 2D Tasks a and b generate a 3D all-task manifold. **b.\** The difference between the dimensionality of all-task and single-task manifolds (Δ in dim) in PMd, M1 and S1 is large, moderate, and small, respectively. The bars show the histogram of single- (grayscale-coded) and all-task (yellow) manifold dimensionality across all sessions for Mk-Ek. The dots show the mean dimensionality of single- (gray) and all-task (yellow) manifolds (single-task: PMd: 3.46 ± 0.92; M1: 3.34 ± 0.48; S1: 2.20 ± 0.46; all-task: PMd: 8.71 ± 2.14; M1: 6.14 ± 0.69; S1: 3.42 ± 0.53). **c** The bars show the mean dimensionality of all-task manifolds across all sessions and both monkeys (PMd: 8.05 ± 0.44; M1: 5.33 ± 0.29 and S1: 4.78 ± 0.44; PMd vs. M1: p = 0.0005; PMd vs. S1: p = 0.0006; M1 vs. S1: p = 0.30; two-sided Wilcoxon signed rank test). Dots show values for each session and monkey (Mk-Ek: left column; Mk-Nt: right column). **d.** Difference (Δ) between dimensionality of all-task and single-task manifolds shown same as in **c** (PMd: 5.81 ± 0.38, M1: 2.66 ± 0.25 and S1: 2.11 ± 0.28; PMd vs. M1: p = 0.00024414; PMd vs. S1: p = 0.0001; M1 vs. S1: p = 0.15; two-sided Wilcoxon signed rank test).

Values shown in (**c**, **d**) are means over 1000 repetitions of randomly selecting 32 channels from each cortical region to avoid bias due to different number of recording channels between regions and monkeys. **e** The panel illustrates the process of computing the alignment index. First, the neural activity of Tasks a and b are used to construct Task a and b single-task manifolds, respectively. Task a neural activity is then projected into the Task b manifold and, separately, into the Task a manifold. To obtain the alignment index, we divided the variance of the projection in Task b manifold by the variance of the projection into Task a manifold. **f** Bar plot shows the alignment index for both monkeys (dots on the left column, Mk-Ek; dots on the right column, Mk-Nt). Smaller alignment index of PMd compared to S1 shows that PMd single-task manifolds differ substantially, while S1 single-task manifolds are similar (n = 14; PMd: 0.38 ± 0.02; M1: 0.47 ± 0.02; S1: 0.56 ± 0.02; PMd vs. M1: p = 0.0005; PMd vs. S1: p = 0.0005; M1 vs. S1: p = 0.0005). Blue line shows the estimated noise contribution with the light blue tube showing the values not significantly different from noise contribution at p ≥ 0.05 (noise contribution: PMd: 0.891 ± 0.009; M1: 0.946 ± 0.005; S1:0.970 ± 0.005; measured value vs. noise contribution: PMd: p = 0.0005; M1: p = 0.0005; S1: p = 0.0005). Error bars: s.e.m.; *** p < 0.001; one-sided Monte Carlo permutation test. Source data are provided as a Source Data file.

the dimensionality of the all-but-one-task manifolds relative to the dimensionality of the all-task manifolds (dimensionality difference: Mk-Ek: PMd: 0.89 ± 0.18, M1: 0.43 ± 0.23, S1: 0.09 ± 0.10; Mk-Nt: PMd: 1.26 ± 0.11, M1: 0.37 ± 0.07; S1: 0.43 ± 0.09; Supplementary Fig. 6e). These relatively small changes indicate that different tasks contributed fairly equally to the dimensionality of the all-task neural manifold.

This result compelled us to quantify the relative similarity between different single-task manifolds in each cortical region, but without the contingence to the dimensionality analysis. For this purpose, we projected the neural activity from one task onto the single-task manifold of another task (Fig. 4e). We then computed the alignment index[14,48], which measures the portion of variance retained after the projection. The alignment index was the largest in S1, significantly smaller in M1, and further significantly smaller in PMd (PMd: 0.38 ± 0.02; M1: 0.47 ± 0.02; S1: 0.56 ± 0.02; PMd vs. M1: p = 0.0005; PMd vs. S1: p = 0.0005; M1 vs. S1: p = 0.0005; one-side Monte Carlo permutation test; Fig. 4f). Taken together, these results show that neuronal activity occurs along neural manifolds that are largely task-specific in PMd, and largely task-agnostic in M1, and even more task-agnostic in S1.

### Task-independent subspace dominates S1 and M1, but not PMd neural activity

Despite the dimensionality of all-task manifolds being larger than single-task manifolds, the sum of all single-task manifold dimensionalities was still larger than the all-task manifold dimensionality. This observation opened up a possibility that, even in PMd, all tasks share activity across one or more neural modes. These modes then span a "task-independent" subspace of the all-task manifold. By construction, the same task-independent subspace is also a subspace within each single-task manifold. Since in PMd the dimensionality of all-task manifolds is substantially larger than the dimensionality of single-task manifolds, we hypothesized that the PMd task-independent subspace holds a smaller portion of neural population activity compared to the

remaining "task-dependent" subspace of the all-task manifold. In contrast, we hypothesized that S1 neural population activity predominantly inhabits the task-independent subspace.

While the dimensionality of the all-task manifold was smaller than the sum of dimensionalities of the single-task manifolds, this quantification does not necessarily demonstrate the presence of the task-independent subspace. For example, two tasks can share some of the neural modes that are not shared across the remaining tasks. In this scenario, the same differences in dimensionalities of single-task and all-task manifolds would emerge even in the absence of a task-independent subspace.

To determine the presence of task-independent subspaces, we applied demixed PCA (dPCA)[30] to the datasets organized based on the peri-gait activity of all multiunits. dPCA separates the neural data into a set of demixed neural modes. In each demixed neural mode, the peri-gait activity of the neural population traces one trajectory for each task. These demixed trajectories can then be decomposed into a task-independent part obtained by averaging the demixed trajectories over the tasks, and a task-dependent part obtained by subtracting the task-independent part from the demixed trajectories (Fig. 5a, Methods). The demixed neural modes are, by the design of the dPCA, dominated by either the task-dependent or the task-independent parts. We hereafter refer to demixed neural modes with majority variance accounted for by the task-independent or task-dependent parts as task-independent or task-dependent modes, respectively. We then defined the task-dependent and task-independent subspaces as spaces spanned by all task-dependent or task-independent modes. As theorized, we identified multi-dimensional task-independent subspaces in all three cortical regions for both monkeys (Fig. 5b, c, and Supplementary Fig. 7).

We next quantified the amount of neural variance captured by task-dependent and task-independent parts of demixed neural modes in each cortical region. A large portion of PMd demixed neural modes were dominated by task-dependent parts, while the majority of S1

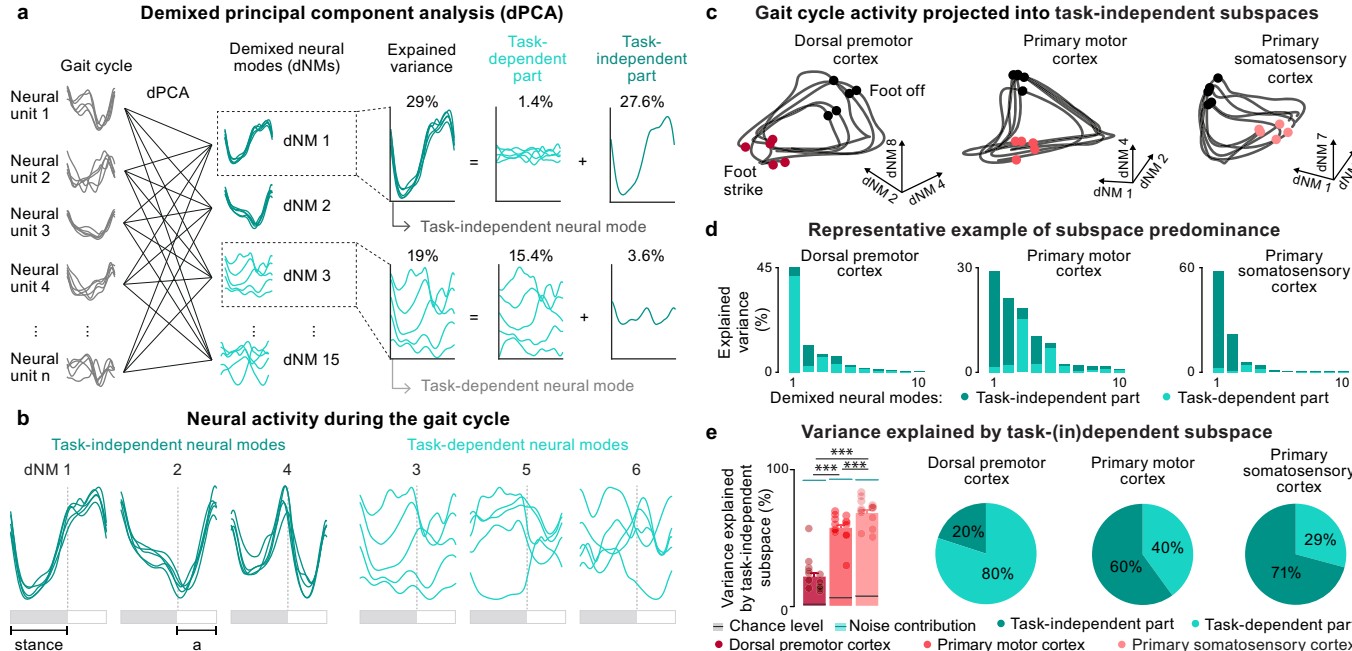

**Fig. 5 | PMd is dominated by task-dependent activity and S1 by task-independent activity. a** We used dPCA to decompose neural population activity of one session into demixed neural modes (dNM), which were then decomposed into task-dependent and task-independent parts. We refer to demixed neural modes as task-dependent or task-independent according to whether the variance is mostly accounted for (>50%) by task-dependent or task-independent parts. **b** Plots show M1 neural population peri-gait activity of three leading task-independent and task-dependent modes for each task in Mk-Ek session S7. All task-independent or task-dependent modes spanned the task-independent or task-dependent subspace respectively. **c** Neural population activity projected into a space spanned by the leading task-independent neural modes. **d** Bars show the portion of the variance explained by each neural mode, and the portion of explained variance belonging to its task-dependent (light green) or task-independent (dark green) parts for the same session as in (**b**). In this representative example in Mk-Ek, variance in the leading PMd demixed neural modes is largely explained by their task-dependent parts. In contrast, variance is largely explained

by the task-independent parts in M1 and S1. **e** Bar plots show the mean portion of variance explained by the task-independent subspace across all sessions and both monkeys (n = 14; S1: 68% ± 3%; M1: 56% ± 3%; PMd: 20% ± 3%; PMd vs. M1: p = 0.0005; PMd vs. S1: p = 0.0005; M1 vs. S1: p = 0.0005). Dots show the value for individual sessions in both monkeys (left column: Mk-EK; right column: Mk-Nt.). Black line shows the chance level with grey tube showing the values not significantly different from chance at p ≥ 0.05 (chance level: PMd: 4.122 ± 0.22; M1: 5.31 ± 0.40; S1: 7.00 ± 0.58; measured value vs. chance: PMd: p = 0.0005; M1: p = 0.0005; S1: p = 0.0005). Blue line shows the estimated noise contribution with light blue tube shows the values not significantly different from noise contribution at p ≥ 0.05 (noise contribution: PMd: 91.73 ± 0.45; M1: 92.55 ± 0.45; S1:91.92 ± 0.45; measured value vs. noise contribution: PMd: p = 0.0005; M1: p = 0.0005; S1: p = 0.0005). Pie charts show the portions of variance explained by task-independent and task-dependent parts of all the demixed neural modes, respectively. Bars: mean ± s.e.m. *** p < 0.001, one-sided Monte Carlo permutation test. Source data are provided as a Source Data file.

modes were captured by task-independent parts (Fig. 5d). Variance of different leading M1 demixed neural modes were dominated by different parts, with more modes being dominated by task-independent parts. In line with these results, M1 and S1 task-independent subspaces captured a substantially larger portion of variance when compared to PMd (S1: 68% ± 3%, M1: 56% ± 3%, PMd: 20% ± 3%; PMd vs. M1: p = 0.0005; PMd vs. S1: p = 0.0005; M1 vs. S1: p = 0.0005; one-sided Monte Carlo permutation test; Fig. 5e). We verified that these differences cannot be explained by sparse task-unrelated behavioral events (Supplementary Fig. 8a). Furthermore, removing leading demixed neural modes from the dataset reduced the variance of all single-task manifolds, thus demonstrating that neural modes did not capture separate task clusters but rather the task-dependent and task-independent activity across all the tasks (Supplementary Fig. 8b). These results add to the evidence that the structure of neural activity is remarkably preserved across locomotor tasks in S1 and to a lesser extent in M1, while PMd primarily exhibits specific neural activity during the performance of each locomotor task.

Despite PMd having four times fewer direct projections to the spinal cord compared to M1[49], the task-specificity of PMd population activity suggests that PMd could drive task-dependent components of muscle contraction or kinematics. To test this interpretation, we used a Wiener filter algorithm to reconstruct the envelopes of EMG activity from the six hindlimb muscles or the right hindlimb kinematics based

on the neuronal activity of PMd, M1, and S1. As expected from earlier studies[50,51], compared to M1 and S1, the PMd neural population activity was less predictive of muscle activity and kinematics (Supplementary Fig. 9a–b). This analysis also showed that, unlike M1 and S1, the PMd neural activity precedes changes in muscle activity and kinematics, which is consistent with the preferential role of PMd in movement planning[11,36] (Supplementary Fig. 9a–b). We then used dPCA to separate the task-dependent and task-independent subspaces of the EMG envelopes (Supplementary Fig. 9c) or the right hindlimb kinematics (Supplementary Fig. 9d). We used the Wiener filter algorithm to reconstruct EMG envelopes and kinematics of these task-dependent and task-independent subspaces from the neural population activity of each of the three cortical regions. As for the decoding of the entire muscle activity and kinematic space, we found that PMd neural population activity provided the worst reconstruction of both task-dependent and task-independent EMG and kinematic subspaces. As for the entire space of EMG and kinematic activity, only the PMd activity preceded the EMG and kinematic subspaces (Supplementary Fig. 9e–h). These results indicate that task-specific neural computations in PMd do not predominantly exert direct control of behavior. Instead, PMd may preferentially pass information to other cortical regions to produce movements. Nonetheless, the possibility that PMd directly controls motor primitives that have not been measured in our experiments cannot be excluded.

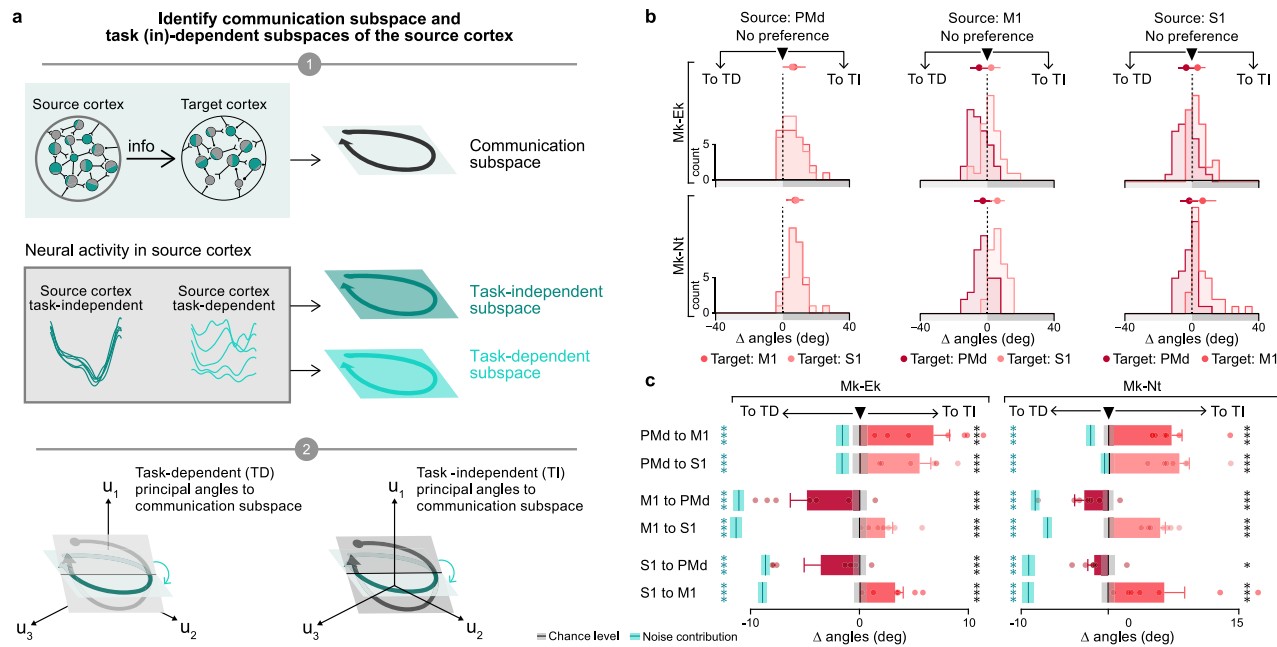

**Fig. 6 | Task-independent subspaces are preferentially aligned to communication subspaces across cortical regions, except those targeting PMd. a Step 1:** We used a reduced rank regression algorithm to identify the source communication subspaces. **Step 2:** We computed the principal angles between the source communication subspaces and source task-independent subspaces (CS-TI angles), as well as the source communication subspaces and source task-dependent subspaces (CS-TD angles). We then computed the angle difference (Δ angles), defined as the difference between the mean CS-TD angles and the mean CS-TI angles. **b** The plots show the distribution of all the Δ angles (n = 35: 5 tasks x 7 sessions) for every combination of source and target cortical regions and for each monkey. Positive Δ angles values show preferential alignment with the task-independent subspaces. **c** Barplots show the mean of Δ angles across all tasks and sessions for every combination of source and target cortical regions and for each monkey (n = 7; Mk-Ek: PMd to M1: 6.80 ± 1.49; PMd to S1: 5.52 ± 1.04; M1 to PMd: -4.82 ± 1.53; M1 to S1: 2.36 ± 0.69; S1 to PMd: −3.55 ± 1.54; S1 to M1: 3.28 ± 0.74; n = 7; Mk-Nt: PMd to M1: 7.28 ± 1.20; PMd to S1: 8.17 ± 1.14; M1 to PMd: −2.75 ± 1.10; M1 to S1: 5.97 ± 0.60; S1 to PMd: −1.62 ± 0.72; S1 to M1: 6.43 ± 2.35). Dots show the mean of Δ angles across all tasks for every session. Black line shows the chance level with the grey tube showing the values not significantly different from chance at p ≥ 0.05 (chance level: Mk-Ek: PMd to M1: 0.05 ± 0.67; PMd to S1: 0.06 ± 0.69; M1 to PMd: 0.03 ± 0.65; M1 to S1:

0.00 ± 0.65; S1 to PMd: 0.04 ± 0.56; S1 to M1: 0.06 ± 0.56; Mk-Nt: PMd to M1: 0.06 ± 0.57; PMd to S1: 0.08 ± 0.57; M1 to PMd: 0.02 ± 0.58; M1 to S1: 0.06 ± 0.56; S1 to PMd: −0.02 ± 0.74; S1 to M1: −0.05 ± 0.75; measured value vs. chance level: Mk-Ek: PMd to M1: p = 0.0005; PMd to S1: p = 0.0005; M1 to PMd: p = 0.0005; M1 to S1: p = 0.0009995; S1 to PMd: p = 0.0005; S1 to M1: p = 0.0005; Mk-Nt: PMd to M1: p = 0.0005; PMd to S1: p = 0.0005; M1 to PMd: p = 0.0005; M1 to S1: p = 0.0005; S1 to PMd: p = 0.02; S1 to M1: p = 0.0005). Blue line shows the estimated noise contribution with the light blue tube showing the values not significantly different from noise contribution at p ≥ 0.05 (noise contribution: Mk-Ek: PMd to M1: −1.56 ± 0.60; PMd to S1: −1.59 ± 0.57; M1 to PMd: −11.07 ± 0.50; M1 to S1: −11.33 ± 0.51; S1 to PMd: −8.63 ± 0.40; S1 to M1: −8.90 ± 0.40; Mk-Nt: PMd to M1: −0.56 ± 0.47; PMd to S1: −0.47 ± 0.44; M1 to PMd: −8.39 ± 0.50; M1 to S1: −6.70 ± 0.48; S1 to PMd: −9.14 ± 0.69; S1 to M1: −9.18 ± 0.68; measured value vs. noise contribution: Mk-Ek: PMd to M1: p = 0.0005; PMd to S1: p = 0.0005; M1 to PMd: p = 0.0005; M1 to S1: p = 0.0005; S1 to PMd: p = 0.0005; S1 to M1: p = 0.0005; Mk-Nt: PMd to M1: p = 0.0005; PMd to S1: p = 0.0005; M1 to PMd: p = 0.0005; M1 to S1: p = 0.0005; S1 to PMd: p = 0.0005; S1 to M1: p = 0.0005). Error bars: s.e.m.; * p < 0.05; *** p < 0.001; one-sided Monte Carlo permutation test. Source data are provided as a Source Data file.

## Communication of task-independent and task-dependent information across cortical regions

We next sought to understand whether each cortical region communicates preferentially task-dependent or task-independent neural activity. Due to the predominance of the task-dependent subspace, PMd may preferentially communicate task-dependent neural activity to the other cortical regions. In contrast, M1 and S1 may preferentially communicate task-independent neural activity, given the predominance of those subspaces in M1 and S1 neural activity.

To test these hypotheses, we applied a reduced rank regression algorithm[52] that aimed to identify the communication subspace of a source cortical region that best explains the activity of a target cortical region[52] for all source-target combinations of the three cortical regions (Fig. 6a, step 1). We then quantified the relative alignment between the source communication subspace and the source task-independent versus task-dependent subspaces by calculating the difference between the mean of the task-dependent principal angles and the mean of the task-independent principal angles[14,53] (Fig. 6a, step 2; for all principal angles, see Supplementary Fig. 10). Thus, a positive or negative value indicated that the source communication subspace is

preferentially aligned to the source task-independent or task-dependent subspace, respectively.

We found that the PMd communication subspaces were preferentially aligned with its less-represented task-independent subspace (mean Δangles: Mk-Ek: PMd to M1: 6.80 ± 1.49; PMd to S1: 5.52 ± 1.04; Mk-Nt: PMd to M1: 7.28 ± 1.20; PMd to S1: 8.17 ± 1.14). In contrast, when considering the communication towards PMd, we found that the communication subspaces from M1 and S1 were preferentially aligned with their less-represented task-dependent subspaces (mean Δangles: Mk-Ek: S1 to PMd: −3.55 ± 1.54; M1 to PMd: −4.82 ± 1.53; Mk-Nt: S1 to PMd: -1.62 ± 0.72; M1 to PMd: -2.75 ± 1.10). Only when communicating between them did the M1 and S1 communication subspaces align with their predominant task-independent subspace (mean Δangles: Mk-Ek: S1 to M1: 3.28 ± 0.74; M1 to S1: 2.36 ± 0.69; Mk-Nt: S1 to M1: 6.43 ± 2.35; M1 to S1: 5.97 ± 0.60; Fig. 6b, c). The preference of PMd-to-M1 communication towards task-independent neural activity suggests that smaller task-specific movement adjustments are planned in PMd. This interpretation is corroborated by the preference of PMd towards receiving task-dependent activity, which would be necessary to perform computation related to task-dependent planning.

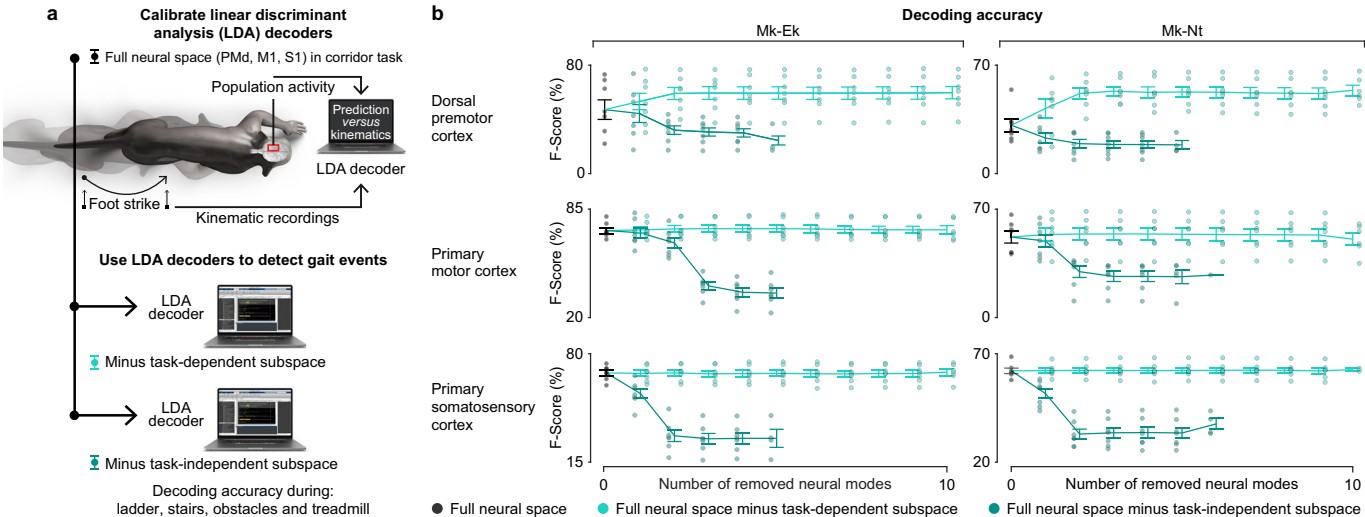

**Fig. 7 | Encoding of gait features in the task-independent subspace remains stable across tasks. a** We used neural decoders to detect right foot-strike and foot-off gait events from neural population activity in the left hemisphere. The decoders were calibrated on the dataset from recordings during walking along the corridor, and were then tested on datasets from the four remaining tasks to evaluate decoding accuracy, as measured by the F-score. We applied this decoding analysis to datasets containing all neural activity from a cortical region (black dot), and to this dataset with one or more leading task-dependent (light green dots) or task-independent (dark green dots) modes removed. **b** Plots show the decoding accuracy for datasets containing all neural activity (black) or the same datasets with leading task-dependent (light green) or task-independent modes (dark green) removed. Dots show data from individual sessions (*n* = 7). Lines show the mean across sessions. Error bars: s.e.m. Source data are provided as a Source Data file.

## Task-independent subspaces enable decoding of temporal features

Task-specific decoding algorithms can accurately infer behavioral features from neural population activity[54,55]. Yet, accuracy drops when decoding behavioral features during tasks that were not used to calibrate the algorithm[54]. The robustness of population activity within the task-independent subspaces suggests that some of the behavioral features are encoded in a portion of the population activity that remains preserved across tasks. Therefore, we hypothesized that decoding accuracy of general behavioral features declines proportionally to the number of available task-independent modes, while removing task-dependent modes has little or no impact on decoding accuracy for these features.

To test this hypothesis, we calibrated a linear discriminant analysis (LDA)[56] decoder to infer the occurrence of right foot-strike and foot-off gait events from neural population activity recorded during walking along the straight corridor. We focused on temporal features, since previous studies suggested that task-independent modes encode the temporal structure of movement[14,34]. Foot-strike and foot-off gait events are key determinants for the production of locomotor movements. Indeed, the prediction of these temporal features from cortical activity drove the electrical stimulation of the spinal cord to restore walking in monkeys[56] and people[57] with paralysis due to spinal cord injury.

We calibrated the decoder under three conditions: full neural space, the full neural space from which we sequentially removed leading task-dependent modes, and the full neural space from which we sequentially removed leading task-independent modes (Fig. 7a). As anticipated, removing task-independent modes drastically reduced the ability of the decoder to predict foot-strike and foot-off gait events during tasks that were not used for calibration. This decline in decoding accuracy was observed for all three cortical regions (Fig. 7b), independently from the amount of removed neural population activity variance (Supplementary Fig. 11d). In striking contrast, removing task-dependent modes did not affect decoding accuracy from S1 and M1, and even slightly increased decoding performance from PMd. These results show that the leading task-dependent PMd modes correlate

with the key temporal features of movement, but that this relationship changes between tasks.

To expand on these findings, we reconstructed EMG envelopes or hindlimb kinematics of non-corridor tasks from neural population activity using a Wiener filter calibrated on the corridor task. Unlike discrete gait events, EMG and kinematics capture the entire range of temporal and spatial features involved in the performance of the tasks. Similar to the detection of gait events, the EMG and kinematic regression accuracy remained unchanged or slightly decreased when removing leading task-dependent components. Yet, removal of leading task-independent components did not consistently reduce regression accuracy (Supplementary Fig. 11b, c). More detailed analysis revealed that the amount of removed variance is more relevant for the regression accuracy reduction than whether the components were task-dependent or task-independent (Supplementary Fig. 11e). These results indicate that the encoding of detailed muscle and kinematic features that generalize across tasks is shared across task-dependent and task-independent subspaces (Fig. 5d).

It is worth mentioning that the robustness of our results is limited by the inability to record all aspects of the monkeys' behavior and environment. While our platform recorded the activity of major hindlimb muscles and movements of primary hindlimb joints synchronously with the activity of the three studied cortical regions, our datasets lacked the activity of the remaining muscles and movements of other joints, some of which are active and relevant during walking. Such more detailed recordings may help explain larger portion of the neural activity variance and, in turn, may lead to higher regression accuracy when reconstructing the motor primitives (Supplementary Fig. 9 and 11). Nonetheless, our key findings do not rely on completeness of behavioral recordings, but simply on the fact that the monkeys performed different tasks, thus allowing us to separate the activity into task-dependent and task-independent components. Despite efforts to standardize conditions, unrecorded sensory inputs (e.g., visual or auditory differences across tasks) may have influenced neural activity, potentially impacting our results.

## Discussion

Here, we took advantage of the naturalistic and rich repertoire of unrestricted primate locomotor behaviors to determine the involvement of multiple cortical regions in the control of movement, and suggest that these principles may be leveraged to increase the reliability of neural prostheses to restore mobility in people with neurological disorders.

We developed a wireless platform that allowed us to monitor neuronal population activity from the hindlimb regions of the PMd, M1, and S1 while the monkeys performed a variety of locomotor tasks under naturalistic conditions. These recordings revealed that the activity of individual neurons from these three cortical regions was highly reproducible within a given task. Accordingly, we found that the activity of neural populations from each region was confined within sets of neural modes that composed low-dimensional manifolds. However, we identified a regional specialization in the changes of the neural population activity that underlies task-specific differences in hindlimb movements. This specialization was captured in the separation of neural population activity into task-dependent and task-independent subspaces. The encoding of motor primitives in S1 changed little between tasks and its activity was dominated by task-independent activity. In contrast, the PMd activity changed substantially between tasks, as reflected by the dominance of task dependent activity. The M1 activity was similar but less pronounced compared to S1: encoding of motor primitives was largely preserved and task-independent activity slightly surpassed the task-dependent activity. Importantly, we found that, despite being dominated by task-independent activity, S1 primarily communicated task-dependent information to PMd. In turn, PMd preferentially communicated its less-prevalent task-independent information to M1. This chain of communication suggests that PMd primarily receives task-dependent information from S1, and then processes this task-dependent information before sending it to M1 in a refined form. Below, we discuss the implications of these fundamental principles for our understanding of the cortical control of movement.

Neurons embedded in the sensorimotor cortices display highly regular patterns of activity that are phase-locked to gait events[54,55,58,59]. Prior studies reported that volitional modifications of locomotor movements to increase speed[54,55,60], walk along uneven terrain[59,61,62], or avoid obstacles[4,63,64] involves task-specific changes of the amplitude and timing of neuronal activity. Here, we expand our understanding of these principles to the multiple regions of the cerebral cortex in primates, showing that the activity of PMd, M1, and S1 neuronal populations is also phase-locked to gait events and displays task-specific changes in the timing and amplitude of activity. However, we found that task-specific changes were not uniformly distributed across the three cortical regions. Overall, S1 showed minimal task dependency, suggesting that neurons located in the hindlimb region of S1 primarily reflect movement primitives[65]. Instead, we detected extensive changes across tasks in the temporal structure underlying the activity of PMd neurons.

PMd is known to be involved in planning, selection, and preparation of goal-directed movements[11,36,66]. Recent studies showed that PMd takes part in executing both internally-generated and externally-guided movements[67,68]. Yet, PMd inactivation predominantly disrupts internally-generated movements, thus indicating that PMd plays a crucial role in the process of converting internal goals into motor commands[68]. Our results shine further light on this process, showing that PMd is focused on neuronal calculations that process contextual task-related information into lower-dimensional motor plans.

There is a widespread agreement that neural populations perform computations through their collective activity[5,15,20,21,24,25,28,69–71]. Indeed, we found that the vast majority of neural population activity from the three studied cortical regions resides within a low-dimensional portion of the full neural space. However, the dimensionality of these neural manifolds increased from S1 (4.78) to M1 (5.33) and again to PMd (8.05). This increase in the complexity of the neural manifolds mirrored the regional specialization of task-specific changes observed in single neuronal responses, which increased from S1 to M1, and further to PMd. Comparable results have been reported during reaching tasks, where the dimensionality of neural manifolds computed from M1 and PMd during reaching attained 8.5 and 14 dimensions, respectively[46]. The increasing dimensionality of neural manifolds with the execution of increasingly more complex behaviors, such as locomotion versus reaching, is in line with the theory that limiting behavior complexity constrains the dimensionality of neural manifolds[47]. In turn, the comparable dimensionalities of forelimb and hindlimb neural manifolds that scale up with behavioral complexity suggest that common principles govern the involvement of cortical neurons in the production of forelimb and hindlimb movements. This interpretation resonates with the viewpoint that, while locomotion and manual dexterity are regarded as separate motor activities, they are intimately connected from an evolutionary and a neurophysiological perspective[3,72]. Indeed, we posit that visuomotor coordination of hand and foot movements, including dexterous prehension, involves comparable neural processes[72].

We found that neural population activity from the three studied cortical regions was compartmentalized into task-dependent and task-independent subspaces. However, we detected a regional specialization in the prevalence of these subspaces. The activity of neural populations from PMd primarily resided in the task-dependent subspace, while the activity of S1 was largely confined within the task-independent subspace. Previous studies suggested that the task-independent subspace conveys the temporal structure of the movement[14,34]. Indeed, we found that decoders based on the task-independent subspaces were able to predict the timing of gait events with remarkable accuracy. Instead, decoders based on task-dependent subspaces failed to establish reliable predictions. We independently validated this concept for the three studied cortical regions. Therefore, our results are not only consistent with previous interpretations on the encoding of the temporal structure of movement in task-independent subspaces, but also expand this principle to multiple regions of the cerebral cortex. The presence of task-independent subspaces across all three cortical regions may coincide with the common cortex-wide dynamics identified in the recent brain-wide calcium imaging recordings in behaving mice[73].

We found that PMd preferentially receives task-dependent information from M1 and S1. In contrast, PMd output to M1 is dominated by task-independent information. This result suggests that PMd transforms task-dependent information from M1 and S1 into task-independent movement plan containing the temporal structure of the ongoing movement. In turn, PMd passes the complete motor plan to M1, now dominated by the more extensive task-independent component. We propose that this principle ensures the coordination between parallel and sequential neural processes that are embedded in distributed regions of the brain, and are mutually involved in the planning and production of movement. These interpretations are aligned with the idea that the brain employs distinct task-specific encoding to convey what should be done and when it should be done[34].

These results have important implications for the design of decoders to operate neural prostheses that remain stable across the varying activities of daily living[74,75]. Decoding algorithms[56,76–78] are generally calibrated from neural activity collected in a given task[54,79]. While these decoders reach excellent levels of performance in the tasks used for decoder calibration, decoding performance degrades when using the decoders in other tasks, which require substantially different patterns of single-neuron activity[74]. Building a decoder that can be applied to a wide variety of tasks requires accounting for, and potentially removing, neural activity that is specific to each context.

Here, we found that decoders based on task-independent subspaces were able to predict the temporal features of hindlimb movements underlying a variety of tasks that encompass the rich repertoire of primate locomotor behaviors. These results indicate that if a sufficiently rich task-independent subspace can be identified, this subspace may be used to build decoders that maintain efficacy during previously-unseen tasks. This concept may be particularly relevant to operate brain-controlled prostheses to regain walking after paralysis, since the control of these prostheses is primarily based on the extraction of the temporal structure of leg movements[56,57,78,80–82].

## Methods

### Statistical procedures

All statistics are reported as mean ± standard error of the mean (s.e.m.). All comparisons between conditions use a null hypothesis of no difference unless stated otherwise.

### Experimental procedures

**Animals.** Two adult female *Macaca fascicularis* monkeys ranging from 7 to 9 years old and weighing between 3.5 and 4.6 kg were involved in this study. Their identifiers are Mk-Ek and Mk-Nt. They were housed in a group in the animal facility of the University of Fribourg (Switzerland) in an enriched indoor room of 45m³ for a group of 2 to 5 animals, as required by Swiss law on animal protection. Additionally, they had access to a 15m³ outdoor space. Animals could interact with other members of the group of 5. They had free access to water and were not food-deprived. All the experimental procedures were approved by the Federal and local veterinary authorities (Service de la sécurité alimentaire et des affaires vétérinaires du canton de Fribourg, authorization numbers: 2016_09_FR and 2019_06_FR).

**Personalized surgery protocol.** We developed a personalized surgical protocol for surgical implantation and fixation of intracortical microelectrode arrays that accounts for NHP-specific anatomical features. We then applied this protocol to obtain long-term simultaneous extracellular recordings of neural populations in PMd, M1, and S1 (Supplementary Fig. 1). We used computer tomography and magnetic resonance imaging to reconstruct and 3D print skull and brain structures. We then co-referenced the reconstructed brain structures to the Paxinos brain atlas[83] to determine the location of the hindlimb areas of PMd, M1 and S1. We 3D-printed the skull and brain from the reconstructions, and then marked the implantation target areas on these prints. Before the surgery, we shaped a titanium mesh to conform to the 3D printed skull and covered it in hydroxyapatite to promote osseointegration. During the surgery, we used the marked 3D print of the skull and the brain to localize the craniotomy and orient the implantation of the microelectrode Utah arrays. After the arrays were implanted and craniotomy was closed, we attached the pedestals holding microelectrode array connectors to footplates on the mesh to secure their position on the head. A more detailed description of the process follows.

**Imaging.** Each animal was lightly sedated with a mixture of ketamine (ketasol-100, 10 mg/kg, intramuscular) and midazolam (Dormicum, 0.1 mg/kg, intramuscular) and brought in a transport cage to the MRI or CT facility of the nearby hospital of Fribourg (HFR). The modalities of animal transport were covered by the veterinary authorization (see above) while the imaging procedures were approved by the authorities of the HFR. At the HFR site, the animal was anaesthetised with a mixture of ketamine (ketasol-100, 4 mg/kg, intramuscular) and medetomidine (Dorbene, 0.04 mg/kg, intramuscular). The animal was then placed in a pronation position in an MRI compatible stereotactic frame (ear bars were covered with Lidohex, a local anaesthetic) and placed in a knee antenna (C-GE-HDx TR Knee PA; G-CoilType=8). An oxygen tube

(3 L/min) was placed in front of the monkey. Latex gloves filled with hot water and a bubble wrap sheet were placed around the animal to prevent temperature drop. Electrocardiogram (ECG) as well as oxygen saturation ($SO_2$) were monitored and recorded every 5 minutes. After the acquisition, the animal received an intramuscular dose of atipamezole (Alzane, 0.2 mg/kg) and was brought back to the animal facility where we closely monitored its condition until complete awakening. Finally, the animal was fed, hydrated, and brought back to the group. The MRI parameters of the acquisition were: (1) acquisition matrix: 256×256; (2) 0.7 mm voxel size; (3) echo time (TE) = 3.3 s; (4) repetition time (TR) = 7.7 s; (5) 3D-T1 and (6) 3D sagittal. The acquisitions were performed on a 3 Tesla GE Medical system (Discovery MR750) machine. The CT parameters of the acquisition were a 0.6 mm helicoidal low dose CT on a Philips Ingenuity TF machine.

The brain structure was extracted using the BET automated brain extraction algorithm[84] in FSL (v.5.0.9) with manual mask removal in FSLView (v.3.2.0). The 3D surface rendering was created in OsiriX (v.3.0.2), cleaned in blender (v.2.78), and down-sampled in Meshmixer (Autodesk, Inc) to be 3D printed (Supplementary Fig. 1a).

### Surgical procedures

**Titanium mesh implantation.** All the surgical procedures described below were performed by a trained functional neurosurgeon in standard sterile conditions. After performing a midline skin incision on the monkey's head, the cranial muscles were separated from the skull. A titanium mesh (TiMesh, Medtronic) previously modeled on the 3D printed skull for each monkey, and covered with hydroxyapatite (Medicoat AG, Zürich, Switzerland) to promote osseointegration, was fixed to the skull with cruciate self-drilling screws (Medtronic, 1.6 × 3.5 mm). On the mesh, two foot-plates covered by a healing plate (Buri SA, La Chaux-de-Fonds, Switzerland) were attached (Supplementary Fig. 1). Finally, the muscles and skin were sutured and closed. After a period to allow for osseointegration (between 8 and 18 weeks), the animal underwent the next surgery. Note that in Mk-Nt, the mesh and chronic intracortical microelectrode array implantations were performed on the same day.

**Chronic intracortical microelectrode array implantation.** After a skin incision, a craniotomy on the left side (~4–5 cm²) was made above the hindlimb areas of dorsal premotor (PMd), primary motor (M1), and primary somatosensory (S1) cortices. The healing plates were removed and a pedestal screwed onto each footplate using eight M1.6 titanium screws (Buri SA, La Chaux-de-Fonds, Switzerland). The dura mater was opened with a surgical blade to expose the brain. Based on anatomical landmarks of the sulci and with the help of the 3D printed brain, chronic microelectrode arrays (UTAH arrays, Blackrock microsystem, USA, 400 μm pitch) were implanted using a pneumatic impactor (Blackrock microsystem, USA). The number of channels varied per animal and cortex: PMd (48 channels in both monkeys; electrode shank length Mk-Ek: 1 mm, Mk-Nt: 1.5 mm), M1 (48 channels in Mk-Ek and 64 channels in Mk-Nt; 1.5 mm electrode shank length), and S1 (64 channels with 1.5 mm electrode shank length in Mk-Ek; and 32 channels with 1 mm electrode shank length in Mk-Nt). Mk-Nt was also implanted with a 48-channel array (1.5 mm electrode tip length) in the posterior parietal cortex, data from this array is not included in this paper. After securing the array, the dura mater was sutured and the craniotome fixed to the skull. Finally, the muscles and skin were sutured for closing.

**Chronic electrode implantation for electromyographic recordings.** Twenty-two silicone-coated, stainless steel electrodes (38 AWG Cooner wires, Omnetics Corporation, USA) were tunneled from the pedestal on the head to the abdomen. A longitudinal skin incision was performed above the targeted muscles and the fascia of each muscle was exposed. Ten electrode pairs were implanted into the muscles after

removing approximately 0.5 mm of the electrodes' insulation. We secured the electrodes and closed the skin. In Mk-Ek, the implanted muscles were: (1) right gluteus medius (GLU); (2) right iliopsoas (IL); (3) right rectus femoris (RF); (4) right semitendinosus (ST); (5) right medial gastrocnemius (MG); (6) right tibialis anterior (TA); (7) right extensor digitorum longus (EDL); (8) right flexor hallucis longus (FHL); (9) left tibialis anterior and (10) left medial gastrocnemius. Two additional electrodes were placed around the back muscles for grounding and reference. Mk-Nt did not undergo EMG implantation.

**Behavioral training.** Animal training was performed based on positive food reinforcement. Initially, the monkeys were trained to come into a custom-made primate chair where we placed reflective markers (either painting or stickers for both hindlimbs) on body landmarks: iliac crest, greater trochanter (hip), lateral condyle (knee), lateral malleolus (ankle) and the 5th metatarsophalangeal (foot). After habituation to the chair, we placed the monkeys in the experimental room for locomotion. We trained the animals to walk on a treadmill (N-mill, Motekforce Link, Netherlands) at different speeds (1–5 km/h). The treadmill was surrounded by a custom-made Plexiglass enclosure (l:146 cm, w:63 cm, h:80 cm) with small openings to give food rewards. After habituation to the treadmill task (more than 30 successive steps), we began overground walking training. The monkeys were placed in a resting box (two custom-made Plexiglass enclosures of l:60 cm, w:40 cm h:72 cm) on either side of the corridor (custom-made 200 cm long, 40 cm wide and 93 cm high). To encourage the monkeys to walk across the corridor (~5–6 steps), we presented a fruit reward on the side of the corridor together with a verbal "go" signal. After the monkeys were accustomed to the task (about 20 crossings), an uneven ladder (10 rungs 2.5 cm in diameter and spaced by 16 cm, 8 cm, 46 cm, 24 cm, 24 cm, 24 cm, 16 cm, 8 cm, 14 cm), one staircase (3 steps up and 3 steps down; w: 35 cm, h:15 cm) and two obstacles (l:40 cm, w: 35 cm, h:20 cm and 30 cm) were introduced. We gradually increased the number of crossings and the number of tasks being performed on the same day. Typically, after 2–4 months, the monkeys were able to achieve all tasks in about 2 hours in the same session. After the training session, we brought the animal back to the animal facility where we gave additional food (primate cereal croquettes) to comply with daily intake requirements.

**Data collection.** During each recording, we acquired hindlimb kinematics with 8 high-resolution cameras (Simi Reality Motion Systems, GmbH, Germany) at 100 Hz, and neural signals at 30 kHz (96-channel CerePlex W and 96-channel CerePlex Exilis, Blackrock Microsystems, USA). Additionally, in Mk-Ek, we acquired hindlimb EMGs at 30 kHz (96-channel CerePlex Exilis, Blackrock Microsystems, USA). Recordings were performed in sessions that each included all five behavioral tasks: corridor, ladder, stairs, obstacles, and treadmill moving at 3 km/h. Each session spanned about 2 hours. Each monkey performed 7 sessions over the period of up to 3 months. Each stairs, obstacles, corridor and ladder recording comprised two back-and-fourth circuit transitions lasting for about 30 seconds. Each treadmill recording involved a minute of steady walking at a constant speed. The acquisition systems were restated and synchronized at the beginning of each recording by sending a sync trigger from the Simi reality motion system to both CerePlex and CerePlex Exilis systems. We randomized the order of tasks in each session.

To make the tasks visually and motivationally similar, the monkeys performed all the tasks in a room without windows and under the same lighting conditions. The monkeys always received the same reward at the end of each crossing of the circuit. The reward provided in the treadmill, where the monkeys stayed in place instead of crossing a circuit, was the same as the reward provided on the same day in each of the other tasks. To minimize the impact of task complexity, we have ensured that all the tasks are easy for the monkeys to perform and that monkeys are highly-trained in performing each of the tasks.

**Data pre-processing**
We marked the right hindlimb foot off and foot strike gait event by visually inspecting the video recordings of the sessions using a custom-made MATLAB program (Mathworks, USA). Gait cycles, defined as the epoch between two consecutive foot strikes, were rejected if the duration of the step was longer than 1.5 seconds. Additionally, we rejected gait cycles with an "outlier" stance phase duration, lasting from the right hindlimb foot strike to the right hindlimb foot off, that exceeded 1.5 times the 75th and 25th interquartile range. In total, there were 295, 312, 124, 208, and 256 gait cycles in Mk-Nt; and 140, 197, 176, 66, and 320 gait cycles in Mk-Ek retained for analysis for corridor, ladder, stairs, obstacles, and treadmill at 3 km/h, respectively. The mean durations of the retained gait cycles were 1056 ms, 988 ms, 1185 ms, 961 ms, and 783 ms for Mk-Nt, and 831 ms, 812 ms, 906 ms, 1127 ms, and 771 ms for Mk-Nt for corridor, ladder, stairs, obstacles, and treadmill at 3 km/h, respectively.

To extract the spike events of isolated neurons, we first concatenated the data streams of all trials of the 5 conditions recorded in a single session. We then performed spike sorting using Offline Sorter (Plexon, Dallas, USA) to identify putative single neurons. Throughout this paper, we refer to these as single neurons.

For neural population activity and multi-dimensional analyses (see below), we identified neural units through threshold crossings (−3.5xRMS) on each electrode. To compute the RMS and threshold crossing events, neural raw data was first cleaned by removing intervals in which one of the channels exceed a threshold value. Cleaned neural data were then filtered between 250 Hz and 5000 Hz and reference to an average value across all channels recording from that cortical region (PMd, M1 or S1). The resulting data stream was used to compute RMS and threshold crossing events. These may include action potentials of well-isolated individual neurons, as well as action potentials of multiple neurons. Throughout this article, we refer to these as multiunits.

For both spikes of single neurons and multiunit spikes, we estimated the instantaneous spike rates by first counting the number of spikes occurring in 10 ms bins in steps of 0.5 ms. We used the binning to calculate the spike rate estimate sampled at 2000Hz. We then convolved these values with a Gaussian kernel (50 ms standard deviation). These firing rates were then Z-scored and down-sampled to 1KHz.

We computed the peri-gait activity of single neurons and multiunits for each task by warping the spike rates to a standard 100-samples-long gait cycle. We first determined that the average duration of the stance phase across tasks to be centered at 60% of the gait cycle duration. We then warped the neural activity during the stance phase into the first 60 samples (i.e., 60% of the gait cycle). The neural activity during the swing phase, covering the remainder of the gait cycle, was warped into the last 40 samples. We performed the warping using the MATLAB function *interp1*. This procedure creates firing rate vectors that can be averaged across steps of different durations. Note that the warping does not change peak firing rate or modulation depth.

All statistics in our study are calculated by treating different sessions as separate data sets with no assumptions about whether the single neurons recorded in different sessions are the same or different.

**Task classification based on EMGs and Kinematic**
To demonstrate the substantial difference in behavior between the five tasks performed by the monkeys, we trained a regularized linear discriminant analysis decoder (rLDA) to classify tasks from EMGs (Mk-Ek) or joint kinematics (Mk-Nt). For each session, we used five-fold cross-validation to obtain five estimates of the decoding accuracy, and then

calculated their mean value. For Mk-Ek, we used EMGs from five right hindlimb muscles and one left hindlimb muscle that showed the most stable recordings in each of the seven analyzed sessions: right iliopsoas (R Il), right rectus femoris (R RF), right medial gastrocnemius (R MG), right extensor digitorum longus (R EDL), right flexor hallucis longus (R FHL) and left tibialis anterior (L TA)). For Mk-Nt, we used x, y, and z coordinates of the hip, knee, ankle, and toe landmarks referenced to the mean position across these four landmarks. We performed the same analysis for each session separately and reported the mean across sessions in Fig. 2e.

### Changes in activity of single neurons across tasks

We identified the preferred gait phase of each single neuron of each session as the sample out of the 100 samples containing the peak average peri-gait activity calculated across the entire session. When generating Fig. 3b, we pooled the single neurons of all sessions and both monkeys together as different neurons and ordered them by their preferred gait phase on each task (top: lowest preferred gait phase). In Fig. 3c, we show the same data with neurons now ordered by their preferred gait phase on the corridor task. Supplementary Fig. 5a shows the same data now separated between the two monkeys. We pooled the data across sessions and monkeys to simplify the presentation, without assumptions whether the single neurons recorded in different sessions are the same or different.

For each neuron and each session, we computed the correlation coefficient ($R$) between peri-gait activity of all possible pairs of gait cycles, including the pairs of gait cycles performed in the same task. Then, we computed the mean of all correlation values for each combination of tasks (5 tasks, 15 combinations). Finally, we derived the cross-task region-specific correlation for each session and task combination by calculating the mean of these values across all neurons of a cortical region. Supplementary Fig. 5a shows the mean cross-task region-specific correlation across sessions. To quantify the mean correlation of the neural population activity across tasks, we calculated the mean of cross-task region-specific correlation across all task combinations excluding the task paired with itself. Bars on Fig. 3d show the mean of these values across all sessions and both monkeys, termed population cross-task correlation, with the dots showing the values for each individual session of each monkey.

We used bootstrapping to evaluate the noise contribution to the population cross-task correlation shown on bars of Fig. 3d. Specifically, for each monkey, session and task, we generated two datasets of neural activity of the same size as the recorded dataset, each by randomly taking the gait cycles with repetitions. Then, for each task, we calculated the correlation across all gait cycle pairs belonging to the two different datasets. These values were then first averaged across all the gait cycle pairs, then across tasks, then across sessions and finally across monkeys. This procedure was then repeated 2000 times to obtain a bootstrap distribution for each cortical region. We calculated the significance of the obtained population cross-task correlation being higher than its bootstrap distribution using a one-sided Monte Carlo permutation test. When comparing the population cross-task correlation between regions, we used a null hypothesis that the difference is equal to the difference of the noise contributions. Specifically, we calculated the significance of the obtained population cross-task correlation differences being higher than the pair-wise difference between their bootstrap distributions using a one-sided Monte Carlo permutation test.

To compare the change in preferred gait phase across tasks for each cortical region, for each neuron and each session, we computed the circular standard deviation of the mean peri-gait single neuron activity across tasks using the Circular Statistics Toolbox for Matlab (see illustration on Fig. 3e). We then computed the mean of these values across all neurons of a cortical region. Bars on Fig. 3e show the mean of these values across all sessions and both monkeys, termed

population PGP circular standard deviation, with the dots showing the values for each individual session of each monkey.

We used bootstrapping to evaluate the noise contribution to the population PGP circular standard deviation shown on bars of Fig. 3e. Specifically, for each monkey, session and task, we generated five datasets of neural activity of the same size as the recorded dataset, each by randomly taking the gait cycles with repetitions. We then calculated the preferred gait phase of each neuron for each monkey, session, task and dataset. We then calculated the circular standard deviation of each neuron across the datasets. These values were then first averaged across all the neurons, then across tasks, then across sessions and finally across monkeys. This procedure was then repeated 2000 times to obtain a bootstrap distribution for each cortical region. We calculated the significance of the obtained population PGP circular standard deviation being higher than its bootstrap distribution using a one-sided Monte Carlo permutation test. When comparing the population PGP circular standard deviation between regions, we used a null hypothesis that the difference is equal to the difference of the noise contributions. Specifically, we calculated the significance of the obtained population PGP circular standard deviation differences being higher than the pair-wise difference between their bootstrap distributions using a one-sided Monte Carlo permutation test.

### Comparison of single-task neural manifolds

To assess whether neural population activity during different tasks were similar, we compared the dimensionality of neural manifolds of single-session datasets containing only one task (single-task manifolds) to the dimensionality of neural manifolds of datasets containing all five tasks (all-task manifolds. We measured the dimensionality by applying PCA on these datasets and then counting the number of leading neural modes needed to explain more than 90% of the cumulative variance[45,46] (data shown in Fig. 4b and Supplementary Fig. 6b). When comparing the dimensionality across cortical regions, to ensure that the results are not biased by the differences in dimensionality of the neural population activity of different regions (i.e., the number of channels recorded from a region), for each session we randomly picked 32 multiunits from regions with a higher channel count and then calculated the single-task and all-task manifold dimensionality. We repeated this process 2000 times. For each session, the reported dimensionality of all-task manifolds (Fig. 4c) and the difference between dimensionality of all-task and single-task manifolds (Fig. 4d) are means over these 2000 repetitions.

To rule out the possibility that the increase in dimensionality was driven by a single task of high dimensionality compared to the others, we performed a control analysis where we computed the dimensionality of the combined-task dataset when a single task was excluded. This method allowed us to understand the contribution of each task towards the increase in dimensionality of the combined dataset (Supplementary Fig. 6e).

We used the alignment index[48] to measure the similarity of orientation of two single-task manifolds. Broadly, the alignment index measures how much of the neural variance of one task can be accounted for when we project its dataset into the single-task neural manifold of a different task (see illustrations on Fig. 4e, f). We calculated the alignment index (AI) in terms of the corresponding reconstruction error as described previously[14,48]:

$$AI = \frac{||X||^2 - ||X - D_m E_m X||^2}{||X||^2} \tag{1}$$

where $X$ is the $n$ by $t$ data matrix containing mean peri-gait neural activity, and matrices $E_m$ and $D_m$ are the encoding and decoding matrices, respectively, derived from the PCA applied on $X$. The matrix $E_m$ projects the original neural data onto the $m$-dimensional neural

manifold, and the matrix $D_m$ optimally reconstructs the original neural data from the latent activity.

For each session and each permutation of two tasks, we calculated the alignment index between the two single-task manifolds (5 tasks, 20 permutations). We then calculated the mean of these alignment indices across all permutations. Bars on Fig. 4f show the mean of these values across all sessions and both monkeys, termed population alignment index, with the dots showing the values for each individual session of each monkey.

We used bootstrapping to evaluate the noise contribution to the population alignment index shown on bars of Fig. 4f. Specifically, for each monkey, session and task, we have generated two datasets of neural activity of the same size as the recorded dataset, each by randomly taking the gait cycles with repetitions. Then, for each monkey, session and task, we calculated the alignment index across the two datasets, i.e., as if the two datasets were two different tasks. These values were then first averaged across tasks, then across sessions and finally across monkeys. This procedure was then repeated 2000 times to obtain a bootstrap distribution for each cortical region. We calculated the significance of the obtained population alignment index of each cortical region being higher than its bootstrap distribution using a one-sided Monte Carlo permutation test. When comparing the population alignment index between regions, we used a null hypothesis that the difference is equal to the difference of the noise contributions. Specifically, we calculated the significance of the obtained population alignment index differences being higher than the pairwise difference between their bootstrap distributions using a one-sided Monte Carlo permutation test.

## Separation of neural activity into task-dependent and task-independent components

We used demixed principal component analysis (dPCA)[30] to distil neural population activity within each cortical region into demixed neural modes that either remained similar across tasks or encoded task-related changes. We referred to these modes as task-independent and task-dependent modes. Each of these modes can be further separated into a task-independent part, obtained by calculating a mean of task-specific neural modes across tasks, averaging population responses for different tasks over the gait cycle, and a task-independent part, obtained by subtracting the task-independent part from the task-specific neural modes.

We begin by concatenating mean-subtracted, trial-averaged, neural data into a 3-dimensional matrix $X$ of size $n$ by $p$ by 100, where $n$ is the number of channels recorded from the considered cortical region, $p$ is the number of tasks, and 100 is the length of the mean peri-gait neural activity for each condition. The matrix $X$ is decomposed into a sum of matrix $X_\theta$, describing behavioral parameters, and matrix $X_{noise}$, describing the measurement noise:

$$X = \Sigma_\theta X_\theta + X_{noise} \qquad (2)$$

Given the decomposition in (2), the loss function of dPCA is given by:

$$L_\theta = ||X_\theta - F_\theta D_\theta X||^2 \qquad (3)$$

where decoder matrix $D_\theta$ and encoder matrix $F_\theta$ are two distinct linear maps, and the activity to be reconstructed is the one demeaned with respect to one of the parameters. In our case, we have only two behavioral parameters $F_\theta$: the gait phase, and the task. After this marginalization, where neural activity is decomposed into parameter-specific averages, it becomes possible to extrapolate the activity related to one parameter by subtracting the average activity of the other parameter. The remaining activity represents the activity related to only one parameter.

## Variance captured by task-dependent and task-independent components of population activity

Using the decomposition in (3), we can split the fraction of explained variance of each dPCA component into the additive contributions of each marginalization[30]. To compute the fraction of variance explained by each dPCA component, we used:

$$R^2 = \sum_\theta \frac{||X_\theta||^2 - ||X_\theta - FDX_\theta||^2}{||X||^2} \qquad (4)$$

The bars on Fig. 5d show how the variance explained by each dPCA component is divided between each marginalization. The pie charts in Fig. 5e show the amount of total variance explained by each marginalization. The bars on Fig. 5e show the mean portion of variance explained by the task-independent subspace across all sessions and both monkeys, termed population task-independent subspace variance, with the dots showing the values for each individual session of each monkey.

We used bootstrapping to evaluate the noise contribution to the population task-independent subspace variance shown on bars of Fig. 5e. Specifically, for each monkey, session, cortical region and task, we generated five datasets of neural activity of the same size as the recorded dataset, each by randomly taking the gait cycles with repetitions. We then identified the task-independent subspaces across the five datasets (i.e., as if the five datasets were five different tasks) for each monkey, session, cortical region and task. We then calculated the portion of variance explained by the task-independent subspaces. These values were then first averaged across tasks, then across sessions and finally across monkeys. This procedure was then repeated 2000 times to obtain a bootstrap distribution for each cortical region. We calculated the significance of the obtained population task-independent subspace variance of each cortical region being higher than its bootstrap distribution using a one-sided Monte Carlo permutation test. When comparing the population task-independent subspace variance between regions, we used a null hypothesis that the difference is equal to the difference of the noise contributions. Specifically, we calculated the significance of the obtained population task-independent subspace variance differences being higher than the pair-wise difference between their bootstrap distributions using a one-sided Monte Carlo permutation test.

## Communication subspace

To define the communication subspace of a source cortical region in relation to a target cortical region during a single session, we used the reduced-rank regression model[52]. We first used a linear model of the form of:

$$Y = XB \qquad (5)$$

where $X$ is a $n$ by $p$ matrix containing the neural population activity of the source cortical region, $Y$ is a $n$ by $q$ matrix containing the neural population activity of the target cortical region, $n$ is the number of gait phase samples of the peri-gait multiunit activity for a single task from a single session (concatenation of time-warped signals), $p$ and $q$ are the number of multiunit channels in the source and target populations, respectively, and $B$ is the coefficient matrix of size $p$ by $q$. Each column of $B$ linearly combines the activity of $p$ channels in $X$ to predict the activity of one column in $Y$. $B$ is calculated using ridge regression:

$$B_{RIDGE} = \left(X^T X + \lambda I\right)^{-1} X^T Y \qquad (6)$$

where $\lambda$ is the regularization coefficient. We determined the value of $\lambda$ using 10-fold cross-validation. For each session separately, we selected $\lambda^*$ as the largest $\lambda$ for which mean performance (across folds), as

measured by $R^2$, the amount of variance explained by the ridge regression, was within one standard error of the mean of the best performance:

$$\hat{Y}_{RIDGE}(\lambda) = XB_{RIDGE}(\lambda) \qquad (7)$$

$$R^2(\lambda) = 1 - \frac{VAR\left(Y - \hat{Y}_{RIDGE}(\lambda)\right)}{VAR(Y)} \qquad (8)$$

Next, we restricted the rank of our model $B$ to $m_{RRR}$. We determined $m_{RRR}$ by first performing principal component analysis of $\hat{Y}_{RIDGE}\left(\lambda^*\right)$. We then generated coefficient matrices $B_{RRR}$ as follows:

$$B_{RRR}(m) = B_{RIDGE}\left(\lambda^*\right)VV^T \qquad (9)$$

where $V$ is a $q$ by $m$ matrix contain the top $m$ principal components of the selected ridge regression predictor $\hat{Y}_{RIDGE}\left(\lambda^*\right)$:

$$\hat{Y}_{RIDGE}\left(\lambda^*\right) = XB_{RIDGE} \qquad (10)$$

We then identified $m_{RRR}$ as the smallest $m$ for which the reconstructed source space activity $\hat{Y}_{RRR}(m) = XB_{RRR}(m)$ explained at least 95% of the cumulative variance reconstructed using ridge regression predictor $\hat{Y}_{RIDGE}\left(\lambda^*\right)$. Across both monkeys and all five tasks, the $m_{RRR}$ ranged from 4 to 32 (mean $m_{RRR}$ for Mk-Ek: $16.7 \pm 0.4$; Mk-Nt: $15.9 \pm 0.6$).

To determine the communication subspace of the source cortical region, we computed:

$$\hat{Y}_{RRR} = XB_{RRR} = XB_{RIDGE}VV^T = X\bar{B}V^T \qquad (11)$$

where $\bar{B} = B_{RIDGE}V$ is a $p$ by $q$ matrix whose columns define which dimensions of the source population activity are used when generating reduced-rank regression predictions. Therefore, the first $m_{RRR}$ columns of $B$ span the communication subspace of the source region.

We quantified the performance of the reduced-rank regression models across different rank $m$ using the coefficient matrices $B_{RRR}(m)$. We then calculated the performance as $R^2$ the portion of target cortex variance explained by the $B_{RRR}(m)$.

$$\hat{Y}_{RRR}(m) = XB_{RRR}(m) \qquad (12)$$

$$R^2(m) = 1 - \frac{VAR\left(Y - \hat{Y}_{RRR}(m)\right)}{VAR(Y)} \qquad (13)$$

Across both monkeys and all five tasks, the $R^2\left(m_{RRR}\right)$ ranged from 0.087 to 0.951 (Mk-Ek: $0.535 \pm 0.017$; Mk-Nt: $0.379 \pm 0.011$).

## Principal angles

Principal angles are a measure of similarity between two linear subspaces[53] by providing an estimate of the linear independence of the two. We computed principal angles between the communication subspace and either the task-dependent or the task-independent subspaces[14]. Since the computation of the principal angles is sensitive to the dimensionality of the subspaces being compared, for this computation we formed the task-dependent and task-independent subspaces from the five leading task-dependent and task-independent neural modes, respectively. This computation provided five task-dependent and task-independent principal angles.

For each monkey, source and target cortical regions, session and task, we calculated the principal angles as the $cos^{-1}$ of the diagonal

elements of matrix $C$ taken from:

$$W_a^T W_b = P_a C P_b^T \qquad (14)$$

where $W_a$ and $W_b$ are the $n$ by $m_a$ and $n$ by $m_b$ matrices that span the communication subspace $a$ and the task-dependent or task-independent subspace $b$; the corresponding neural modes are their column vectors, and $P_a$ of size $m_a$ by $m_b$ and $P_b$ of size $m_b$ by $m_b$ are the new bases of the low-dimensional subspaces minimizing the principal angles. We then calculated the Δangles as the difference between the mean of the task-dependent principal angles and the mean of the task-independent principal angles. This gave us one value for each monkey, source-target pairing of cortical regions, session and task. Figure 6b shows the distribution of all the 35 obtained values (7 sessions x 5 tasks) for each source-target pairing of cortical regions for each monkey. Bars on Fig. 6c show the mean of these values across all tasks and all sessions, with the dots showing the values for each individual session.

We evaluated the noise contribution to the mean Δangles shown on bars of Fig. 6c as follows. For each source-target pairing of cortical regions, we broke down the temporal correlations between the source and the target cortex, and within the target cortex itself. We did so by advancing the neural activity of each gait cycle and each channel of the target cortex dataset by a different gait phase randomly drawn from a uniform distribution. We then computed the communication subspace using the "intact" source cortical region and the "shuffled" target cortical region. We then calculated the principal angles between this communication subspace and the task-dependent and task-independent subspaces of the intact source cortical region, and computed the Δangles. This process was done for each session and task separately. We then used these 35 values to obtain the mean Δangles. We repeated this process 2000 times to obtain the shuffling distribution. We reported the p value of the measured mean Δangles being higher than this bootstrap distribution using a one-sided Monte Carlo permutation test. The mean and standard deviation of the bootstrap distribution are now shown on Fig. 6c as the blue line and blue region, respectively.

## Decoding of gait events and generalization across tasks

We evaluated the performance of decoders calibrated on the neural population activity and gait event data of the corridor task and then applied on the neural population activity of one the other four tasks to detect the gait events. These decoders used the complete neural population activity of one of the cortical regions, or the neural population activity with one or more leading task-dependent or task-independent neural modes removed. We then compared the performance of these decoders to determine which features of locomotor behavior are encoded in the task-dependent and task-independent subspaces of the neural activity in each of the cortical regions.

To detect gait events, we used a multiclass regularized linear discriminant analysis (rLDA) decoding model as previously described[56,82,85]. Briefly, to calibrate the decoder, we used the right hindlimb foot-strike and foot-off gait events to identify sets of neuronal features used to calibrate the decoders. We derived three motor state classes of neural features based on these two gait events and "no event" periods between the two gait events. The amount of neural data taken before each gait event (feature length), the number of bins within it (feature dimension), and the regularization coefficient for each decoder were determined by cross-validation[56,82,85]. During the use of the decoder, when one of the gait event (foot-strike or foot-off) probabilities crossed an 80% threshold, that event was 'detected'. There was a refractory period preventing a detection of the same gait event during 100 ms. Once we determined the decoder parameters, we calibrated the decoder on neural features derived from the entire

corridor dataset. We then applied the decoder to neural population activity in other tasks of the same session to detect the gait events in that dataset. We quantified the decoding performance using the mean F-score, as defined as the harmonic average of per-class recall and precision, across the four non-corridor tasks. Plots in Fig. 7b and Supplementary Fig. 10d show the mean of this F-score across all sessions.

## Wiener filter reconstruction of EMGs and kinematics

We used a Wiener cascade filter[86] to reconstruct kinematic and EMG time series from neural population activity. In Mk-Ek, we used the filter to reconstruct EMG envelopes of right IL, RF, MG, EDL and FHL; and left TA. In Mk-Nt, we used the filter to infer x, y and z velocities of right hip, knee, ankle, and toe joints referenced to the mean position across these four landmarks. We used Savitsky-golay filter to estimate the velocities out of the recorded joint positions. We evaluated the performance of the decoder using the R2 metric between the reconstructed and true EMG envelopes (Mk-Ek) or joint velocities (Mk-Nt). The Wiener filter used a 100 ms and 300 ms window of neural population activity to reconstruct EMG envelopes and joint velocities, respectively. To calibrate the Wiener filter, for each session and cortical region, we first used five-fold cross-validation during the corridor tasks to determine the lambda parameter of the filter. We then calibrated the filter using the determined lambda on the entire corridor dataset, and applied it to the neural population activity in other tasks of the same session to reconstruct the EMG envelopes or joint velocities from the neural population activity. Finally, we calculated the mean $R^2$ across all four non-corridor tasks. We then repeated this process for neural population activity with one or more leading task-dependent or task-independent neural modes removed. Plots in Supplementary Fig. 10b,c,e show the mean of the final $R^2$ across all sessions.

## Chance level estimation

We estimated the chance level within a single condition for the analysis in Figs. 3d and 5e by advancing the neural activity warped trials for each channel and each gait cycle by a different gait phase randomly drawn from a uniform distribution, and then calculating the respective statistic. We repeated this process 2000 times per analysis. We calculated the chance level in the Fig. 3e by randomly drawing five angles from a uniform distribution, and then calculating the circular standard deviation. We calculated the chance level in the Fig. 6c by calculating the principal angles between either the task-dependent or task-independent subspaces spanned by the five leading neural modes and, a randomly generated $n$-dimensional subspace ($n$ = the number of the communication subspace dimensions). We repeated this process 2000 times per analysis.

## Reporting summary

Further information on research design is available in the Nature Portfolio Reporting Summary linked to this article.

## Data availability

Source data are provided with this paper.

## Code availability

Software routines developed for the data analysis will be made available upon request to the corresponding authors.

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

## Acknowledgements
Defitech Foundation, the Wyss Center for Bio- and Neuroengineering, Bertarelli Foundation, a Whitaker foundation fellowship to M.G.P. and the Swiss National Science Foundation. (CRSII5_183519).

## Author contributions
Technological framework: S.B., N.M., A.M.H, M.G.P., F.F., S.P.L., I.S. Surgeries: M.G.P., S.B., E.M.R., G.C., J.B. Performed experiments: S.B., N.M., A.M.H, M.G.P., C.H., I.S., M.D. and R.O.K. Data analysis: S.B., N.M., M.G.P., H.J., I.S, T.M. Conception and supervision: S.P.L., E.B., E.M.R., J.B., T.M., I.S., G.C. T.M. wrote the paper with S.B., N.M., I.S., M.G.P. and G.C.

## Competing interests
Authors declare no competing interests related to presented work.

## Additional information

[1]NeuroX Institute and Brain Mind Institute, School of Life Sciences, Swiss Federal Institute of Technology (EPFL), Lausanne, Switzerland. [2]Department of Clinical Neuroscience, Lausanne University Hospital (CHUV) and University of Lausanne (UNIL), Lausanne, Switzerland. [3]NeuroRestore, EPFL/CHUV/UNIL, Lausanne, Switzerland. [4]Department of Neurosciences and Movement Sciences, University of Fribourg, Fribourg, Switzerland. [5]Center for the Neural Basis of Cognition, Department of Bioengineering, University of Pittsburgh, Pittsburgh, PA, USA. [6]Rehab Neural Engineering Labs, Department of Physical Medicine and Rehabilitation, University of Pittsburgh, Pittsburgh, PA, USA. [7]Department of Fundamental Neuroscience, University of Geneva, Geneva, Switzerland. [8]Department of Neuroscience, Icahn School of Medicine at Mount Sinai, New York, NY, USA. [9]Institute of Neuroinformatics, ETH Zürich and University of Zürich, Zürich, Switzerland. [10]Bertarelli Foundation Chair in Neuroprosthetic Technology, Institute of Microengineering, Institute of Bioengineering, NeuroX Institute, Swiss Federal Institute of Technology (EPFL), Lausanne, Switzerland. [11]University of Bordeaux, CNRS, IMN, UMR 5293, Bordeaux, France. [12]Department of Biomedical Engineering, Washington University in St. Louis, St. Louis, MO, USA. [13]Department of Neurosurgery, Washington University School of Medicine, St. Louis, MO, USA. [14]Division of Neurotechnology, Washington University School of Medicine, St. Louis, MO, USA. [15]These authors contributed equally: Simon Borgognon, Nicolò Macellari. [16]These authors jointly supervised this work: Tomislav Milekovic, Ismael Seáñez, Grégoire Courtine. ✉e-mail: tomislav.milekovic@epfl.ch; ismaelseanez@wustl.edu; gregoire.courtine@epfl.ch

