## [Transparent Peer Review file · Nature Communications]

Regional specialization of movement encoding across the primate sensorimotor cortex

Corresponding Author: Professor Gregoire Courtine

This manuscript has been previously reviewed at another journal. This document only contains information relating to versions considered at Nature Communications. Mentions of the other journal have been redacted.

Version 0:

Reviewer comments:

Reviewer #1

(Remarks to the Author)
Review #3 of Borgonon et al.

I find the revised manuscript substantially improved, and my concerns have largely been addressed. I have just a few minor concerns that remain:

Page 8: "Excluding one task from the calculation of the neural manifold had no significant impact on the dimensionality of the all-task neural manifold (Supp. Fig. 6e). This invariance indicated that different tasks contributed fairly equally to the dimensionality of the all-task neural manifold." There is no foundation for concluding from a failed statistical test that quantities are "fairly equal," nor does the characterization of the result as an "invariance" (here and in the next paragraph) have a clear foundation in the quantification provided. If the authors want to state that the quantities in question are fairly equal, they can measure the percent difference between the quantities. I understand that the misuse of statistics is endemic in contemporary literature, but that does not make it acceptable.

Page 9: "Despite the limited number of direct projections from PMd to the spinal cord [49]" - This statement is not accurate, and it is not supported by the reference provided. The thrust of the two papers by He et al. was quite the opposite. Here is a relevant passage from the companion paper to the one referenced, which analyzes the same dataset (He et al., 1993, JNsci, 13(3): 952:980):

"First, the PMd has substantial projections to the spinal cord. Previously, it had been suggested that an important difference between the PMd and the primary motor cortex was that corticospinal neurons were largely absent from the PMd (e.g., Sessle and Wiesendanger, 1982). Our results do not support this suggestion, and instead confirm our prior observation that the density of corticospinal neurons in the PMd is comparable to that in the primary motor cortex (see also Dum and Strick, 1991b). This result has led us to view the PMd as one of a set of cortical areas that, like the primary motor cortex, has the potential to influence directly the generation and control of movement at the level of the spinal cord (Dum and Strick, 1991a,b)."

It is true, as the authors of the present manuscript have said in their rebuttal, that the data reported in these papers indicate ~4x more corticospinals in M1 that target both lumbrosacral and cervical segments, but one also has to account for the fact that PMd is half the size of M1, as defined in He et al.. Characterizing this as indicating a "limited number of direct projections from PMd," gives the reader the wrong impression.

Figure 6 legend: given the mass of p-values added, to aid readability I would suggest these instead be included in a supplementary table.

Page 11: "likely reflecting the fact that task-dependent components contain some task-independent-parts (Fig. 5d)." I do not agree with this statement, as it ignores the possibility that descending commands themselves can be task-dependent. Particularly in the context of BCI-related work, the likelihood of this possibility is important and I think should not be ignored.

After addressing these minor issues, I think the manuscript should be accepted for publication.

Reviewer #2

(Remarks to the Author)

My remaining concerns have been addressed, and the synthesis is significantly improved. Note that there is a typo in some new text: on L390, I believe “independent” should read “dependent”.

Reviewer #3

(Remarks to the Author)

Regional specialization of task-specific movement encoding across the primate sensorimotor cortex by Borgognon et al

This is a re-review of this manuscript that was transferred from [REDACTED]. I continue to feel that this is an interesting dataset that will be of interest for the field. The dataset is rich, recording neural activity from three different brain regions as animals perform five different locomotion tasks. The analyses are cutting-edge and appropriate for the questions at hand.

In the revised manuscript, the authors have addressed my remaining concerns:

1. They noted the potential confound of differences in sensory representations explaining the differences in task-dependent and task-independent representations across regions. This is included right before the Discussion. I would have liked to see this more clearly explained, but it is there for the reader to interpret as they see fit.
2. The authors have added statistics, as requested.
3. The authors provide an estimate of the noise floor for some of the cross-task comparisons. In particular, Figures 3d, 3e, 4f, 5e, and 6c all include noise floors. I thank the reviewer's for considering my suggestion and I think this improves the interpretability of these results. One small comment remains – it remains unclear to me whether the ordering of neural activity in Figures 3b and 3c are also done in a split-half manner? I apologize if I missed where that was described (didn't see it in either the figure legend or methods). Ideally, these plots would also be cross-validated across split-halves.
4. I thank the authors for including the absolute principal angles between different subspaces in Supplementary Figure 10. This clarifies the overall alignment between subspaces and the effect of the difference in angle. I would make the minor suggestion of explicitly stating that the angle is included in Supp. Fig. 10 in the main manuscript (it took me a while to find it).

AUTHORS' ANSWER: Answer to each query.

MANUSCRIPT MODIFICATION: Description of actions taken in the manuscript, with changes marked in **bold letters**.

REVIEWERS' COMMENTS

Reviewer #1 (Remarks to the Author):

Review #3 of Borgonon et al.

I find the revised manuscript substantially improved, and my concerns have largely been addressed. I have just a few minor concerns that remain:

Page 8: "Excluding one task from the calculation of the neural manifold had no significant impact on the dimensionality of the all-task neural manifold (Supp. Fig. 6e). This invariance indicated that different tasks contributed fairly equally to the dimensionality of the all-task neural manifold." There is no foundation for concluding from a failed statistical test that quantities are "fairly equal," nor does the characterization of the result as an "invariance" (here and in the next paragraph) have a clear foundation in the quantification provided. If the authors want to state that the quantities in question are fairly equal, they can measure the percent difference between the quantities. I understand that the misuse of statistics is endemic in contemporary literature, but that does not make it acceptable.

AUTHORS' ANSWER: We thank the reviewer for emphasizing the importance of precise language when interpreting results. We have now revised the text to specifically provide the mean and s.e.m. of the differences in dimensionalities between all-task and all-but-one-task manifolds across both monkeys and all three cortical regions. We have also removed any reference to the term "invariance".

MANUSCRIPT MODIFICATION: We have changed the sentences referenced by the reviewer to state:

"Excluding one task from the **datasets modestly reduced the dimensionality of the all-but-one-task manifolds relative to the dimensionality of the all-task manifolds (dimensionality difference: Mk-Ek: PMd: 0.89 ± 0.18 , M1: 0.43 ± 0.23 , S1: 0.09 ± 0.10 ; Mk-Nt: PMd: 1.26 ± 0.11 , M1: 0.37 ± 0.07 ; S1: 0.43 ± 0.09 ; Supp. Fig. 6e). These relatively small changes** indicate that different tasks contributed fairly equally to the dimensionality of the all-task neural manifold.

This **result** compelled us to quantify the relative similarity between different single-task manifolds in each cortical region, but without the contingency to the dimensionality analysis."

Page 9: "Despite the limited number of direct projections from PMd to the spinal cord [49]" - This statement is not accurate, and it is not supported by the reference provided. The thrust of the two papers by He et al. was quite the opposite. Here is a relevant passage from the companion paper to the one referenced, which analyzes the same dataset (He et al., 1993, JNsci, 13(3): 952:980):

"First, the PMd has substantial projections to the spinal cord. Previously, it had been suggested that an important difference between the PMd and the primary motor cortex was that corticospinal neurons were largely absent from the PMd (e.g., Sessle and Wiesendanger, 1982). Our results do not support this suggestion, and instead confirm our prior observation that the density of corticospinal neurons in the PMd is comparable to that in the primary motor cortex (see also Dum and Strick, 1991b). This result has led us to view the PMd as one of a set of cortical areas that, like the primary motor cortex, has the potential to influence directly the generation and control of movement at the level of the spinal cord (Dum and Strick, 1991a,b)."

It is true, as the authors of the present manuscript have said in their rebuttal, that the data reported in these papers indicate ~4x more corticospinals in M1 that target both lumbrosacral and cervical segments, but one also has to

account for the fact that PMd is half the size of M1, as defined in He et al.. Characterizing this as indicating a “limited number of direct projections from PMd,” gives the reader the wrong impression.

AUTHORS’ ANSWER: We thank the reviewer for the thoughtful and well-supported correction. We agree that our original phrasing overstated the contrast between PMd and M1 projections to the spinal cord and did not accurately reflect the conclusions of He et al. In particular, we appreciate the clarification that, while PMd has fewer corticospinal neurons compared to M1, its projection density is still substantial. We have now changed the manuscript text accordingly.

MANUSCRIPT MODIFICATION: We have modified the relevant sentence to state:

“Despite **PMd having four times fewer direct projections to the spinal cord compared to M1⁴⁹**, the task-specificity of PMd population activity suggests that PMd could drive task-dependent components of muscle contraction or kinematics.”

Figure 6 legend: given the mass of p-values added, to aid readability I would suggest these instead be included in a supplementary table.

AUTHORS’ ANSWER: The exact p-values were added to the captions of the figures on request of Reviewer 3 in order to comply with the Nature guidelines on presenting statistics.

Page 11: “likely reflecting the fact that task-dependent components contain some task-independent-parts (Fig. 5d).” I do not agree with this statement, as it ignores the possibility that descending commands themselves can be task-dependent. Particularly in the context of BCI-related work, the likelihood of this possibility is important and I think should not be ignored.

AUTHORS’ ANSWER: We have now deleted the part of the sentence referred to by the reviewer.

MANUSCRIPT MODIFICATION: We have modified the relevant sentence to state:

“These results indicate that the encoding of detailed muscle and kinematic features that generalize across tasks is shared across task-dependent and task-independent subspaces (Fig. 5d).”

After addressing these minor issues, I think the manuscript should be accepted for publication.

AUTHORS’ ANSWER: We thank the reviewer for their constructive feedback that helped improve the clarity and accuracy of our work.

Reviewer #2 (Remarks to the Author):

My remaining concerns have been addressed, and the synthesis is significantly improved. Note that there is a typo in some new text: on L390, I believe "independent" should read "dependent".

AUTHORS' ANSWER: We sincerely thank the reviewer for their positive feedback and are pleased to hear that the revised manuscript addresses the remaining concerns and is viewed as significantly improved.

MANUSCRIPT MODIFICATION: We have corrected the typo found by the reviewer. The referenced sentence now states:

"In contrast, the PMd activity changed substantially between tasks, as reflected by the dominance of task dependent activity."

Reviewer #3 (Remarks to the Author):

Regional specialization of task-specific movement encoding across the primate sensorimotor cortex by Borgognon et al

This is a re-review of this manuscript that was transferred from [REDACTED]. I continue to feel that this is an interesting dataset that will be of interest for the field. The dataset is rich, recording neural activity from three different brain regions as animals perform five different locomotion tasks. The analyses are cutting-edge and appropriate for the questions at hand.

In the revised manuscript, the authors have addressed my remaining concerns:

1. They noted the potential confound of differences in sensory representations explaining the differences in task-dependent and task-independent representations across regions. This is included right before the Discussion. I would have liked to see this more clearly explained, but it is there for the reader to interpret as they see fit.

AUTHORS' ANSWER: We thank the reviewer for their efforts to improve the clarity of our article. We have now slightly rephrased this section to clarify it.

MANUSCRIPT MODIFICATION: The referenced paragraph now states:

“It is worth mentioning that the robustness of our results is limited by the inability to record all aspects of the monkeys' behavior and environment. While our platform recorded the activity of major **hindlimb** muscles and movements of primary **hindlimb** joints synchronously with the activity of the three studied cortical regions, our datasets lacked the activity of the remaining muscles and movements of other joints, some of which are active and relevant during walking. **Such** more detailed recordings may help explain larger portion of the neural activity variance and, in turn, **may** lead to higher regression accuracy when reconstructing the motor primitives (Supp. Fig. 9 and 11). Nonetheless, our key findings do not rely on completeness of behavioral recordings, but simply on the fact that the monkeys performed different tasks, thus allowing us to separate the activity into task-dependent and task-independent components. Despite efforts to standardize conditions, unrecorded sensory inputs (e.g., visual or auditory differences across tasks) may have influenced neural activity, potentially impacting our results. “

2. The authors have added statistics, as requested.

AUTHORS' ANSWER: We thank the reviewer for their help in bringing our manuscript up to the Nature Communications standards for reporting statistical data.

3. The authors provide an estimate of the noise floor for some of the cross-task comparisons. In particular, Figures 3d, 3e, 4f, 5e, and 6c all include noise floors. I thank the reviewer's for considering my suggestion and I think this improves the interpretability of these results. One small comment remains – it remains unclear to me whether the ordering of neural activity in Figures 3b and 3c are also done in a split-half manner? I apologize if I missed where that was described (didn't see it in either the figure legend or methods). Ideally, these plots would also be cross-validated across split-halves.

AUTHORS' ANSWER: We thank the reviewer for pointing out our lack of clarity. To improve transparency, we have now modified the relevant section in the methods that explains how we generated these figures.

MANUSCRIPT MODIFICATION: The relevant sentence now states:

“We identified the preferred gait phase of each single neuron of each session as the sample **out of the 100 samples** containing the peak average peri-gait activity **calculated across the entire session**. When generating Fig. 3b, we pooled the single neurons of all sessions and both monkeys together as different neurons and ordered them by their

preferred gait phase on each task (top: lowest preferred gait phase). In Fig. 3c, we show the same data with neurons now ordered by their preferred gait phase on the corridor task.”

4. I thank the authors for including the absolute principal angles between different subspaces in Supplementary Figure 10. This clarifies the overall alignment between subspaces and the effect of the difference in angle. I would make the minor suggestion of explicitly stating that the angle is included in Supp. Fig. 10 in the main manuscript (it took me a while to find it).

AUTHORS' ANSWER: Many thanks to the reviewer for this helpful comment. We have now included the manuscript text pointing the reader to Supp. Fig. 10 for specific values of the principal angles.

MANUSCRIPT MODIFICATION: The relevant sentence now states:

“We then quantified the relative alignment between the source communication subspace and the source task-independent versus task-dependent subspaces by calculating the difference between the mean of the task-dependent principal angles and the mean of the task-independent principal angles^{14, 53} (Fig. 6a, step 2; **for all principal angles, see** Supp. Fig. 10).”